# Wind inflow observation from load harmonics: initial steps towards a field validation

Marta Bertelè[1], Carlo L. Bottasso[1], and Johannes Schreiber[1]

[1]Wind Energy Institute, Technische Universität München, Garching bei München, D-85748 Germany

*Correspondence to:* C.L. Bottasso (carlo.bottasso@tum.de)

**Abstract.**

A previously published wind sensing method is applied to an experimental dataset obtained on a 3.5 MW turbine. The method is based on a load-wind model that correlates once-per-revolution blade load harmonics to rotor-equivalent shears and wind directions. Loads measured during turbine operation are used to estimate online —through the load-wind model— the inflow at the rotor disk, thereby turning the whole turbine into a sort of generalized anemometer.

The experimental dataset consists of synchronous measurements of loads, from blade-mounted strain gages, and of the inflow, obtained from a nearby met mast. As the mast reaches only to hub height, a second independent method is used to extend the met-mast-measured shear above hub height to cover the entire rotor disk. Part of the dataset is first used to identify the load-wind model, and then the performance of the wind observer is characterized with the rest of the data.

Although the experimental setup falls short of providing a thorough validation of the method, it still allows for a realistic practical demonstration of some of its main features. Results indicate a good quality of the estimated linear shear, both in terms of 1 and 10-min averages and of resolved time histories, with mean average errors around 0.04. A similarly accuracy is found in the estimation of the yaw misalignment, with mean errors typically below 3 deg.

## 1   Introduction

This paper presents a first attempt at the field validation of a wind sensing method based on load harmonics. Wind sensing refers to the general concept of using the response of the turbine to estimate certain characteristics of the inflow, a task that can be accomplished in several different ways (Bottasso et al., 2010; Bottasso and Riboldi, 2014; Simley and Pao, 2016; Bottasso and Riboldi, 2015; Bertelè et al., 2017; Bottasso et al., 2018; Schreiber et al., 2020). Information on the inflow can support a variety of applications, including turbine and farm-level control, lifetime assessment and fatigue consumption estimation, power and wind forecasting, and others (Schreiber et al., 2020).

A detailed knowledge of the wind inflow is today lacking in essentially all wind turbine installations, with the exception of experimental prototypes, certification tests and turbines equipped with forwarding looking lidars (Scholbrock et al., 2013; Schlipf et al., 2014; Peña Diaz et al., 2014). In fact, standard production machines are typically equipped with nacelle-mounted anemometers and wind vanes. These sensors need to be carefully calibrated to eliminate effects caused by —among others— the rotor wake, blade passage, and flow distortions caused by the large bluff body represented by the nacelle. Even when these

effects are properly accounted for, nacelle mounted sensors suffer from the unavoidable limitation of providing only point-wise measurements. As such, they are blind to flow features that are characterized by a variability of the wind field over the rotor swept area, namely horizontal and vertical shears, veer, or the presence of an impinging wake released by a turbine located upstream. Additionally, especially for today's very large turbines of ever increasing diameters, point-wise measurements might

not fully reflect the actual conditions experienced at the rotor disk. At some wind plants, met masts are available and can in principle provide additional information on the wind characteristics at different heights above ground. However, here again this information is of limited use: only a small number of masts is typically available at production sites and, clearly, these measurements are not co-located with the turbines. Lidars and radars (Lang and McKeogh, 2011; Scholbrock et al., 2013; Mikkelsen, 2014; Schlipf et al., 2014; Hirth et al., 2015; Valldecabres et al., 2018; Peña Diaz et al., 2014) are remote sensing

devices that can be used to scan the flow, providing maps of wind characteristics in space and time. Such devices are however not yet routinely used on production machines, because of cost, reliability and availability issues.

Wind sensing was first proposed to address the need for a simple, low-cost way of measuring the inflow at the rotor disk during operation of a turbine, a capability that is today still lacking.

The wind sensing approach is based on two main observations.

The first observation is mainly an economic one, and relates to the opportunity offered by sensor technology, in particular strain gages, accelerometers and pressure sensors. While sensors have been and still are routinely used on prototypes and during certification and research tests, they are today becoming more commonly deployed also on production machines, for example for enabling load-reducing control, or for condition monitoring, digital twin applications, fault-icing-erosion detection etc. Indeed, a growing number of use cases, improved technology and reduced purchase and maintenance costs have led

several OEMs to equip their latest models with rotor sensors, while retrofitting solutions are becoming readily available on the market (Bachmann, 2021; fos4X, 2021; Wind-Consult, 2021). The question is then: can these same devices also be used for wind sensing? In principle, a positive answer to this question opens up very interesting opportunities: when a rotor is already sensorized, the estimation of the wind inflow could be a simple software upgrade, with no extra equipment needed, and therefore no extra purchase and maintenance costs. One could then, at no or very limited cost, turn each rotor in a farm into

a sophisticated anemometer, this way providing a wealth of information on the actual flow conditions throughout the plant. But even if a rotor is not already equipped with sensors, a wind sensing technology based on existing proven sensors readily available on the market might still be very attractive and economically viable. Based on this first observation, the development of wind sensing has mainly revolved up to now on the idea of using information provided by blade load sensors (either standard bridge-based ones, or optical ones based on Fiber Bragg Grating technology (Schubel et al., 2013)), which is also the approach

pursued in this paper. However, future work could try to exploit accelerometers, alone or fused together with load sensors.

The second observation at the heart of wind sensing is the rather obvious one that changes in the wind inflow will affect the rotor response. The basic scientific question is then: if one could measure the rotor response (through the rotor sensors discussed above), would it be possible to infer the wind inflow from such measurements? In other words, is there a wind-response relationship that can be inverted to estimate the inflow given a measurement of the response? A positive answer

to this question was first given by Bottasso and Riboldi (2014), which showed that shear and misalignment (i.e., a relative

angle between rotor axis and wind vector) do leave distinguishing effects on the once-per-revolution (1P) sine and cosine harmonic components of the blade response. The method, termed here *harmonic-based* approach, was then further elaborated and improved by Bottasso and Riboldi (2015); Cacciola et al. (2016a); Bertelè et al. (2017, 2018, 2019). With time, a simpler method was developed (Bottasso et al., 2018; Schreiber et al., 2016, 2020), which uses blade load time histories to estimate the average wind over sectors of the rotor disk. This second method, termed here *sector-effective* approach to distinguish it from the harmonic-based one, is capable of detecting shears and an impinging wake (Schreiber et al., 2020), but not wind directions.

The present paper uses an existing dataset, previously collected for other purposes, to attempt a first field validation of the harmonic-based wind inflow estimator. In short, this method is based on a correlation between the in- and out-of-plane blade root bending 1P harmonics and four specific characteristics of the rotor inflow: namely, the horizontal and vertical shears, and the lateral and vertical misalignment angles. Indeed, it can be shown (Bertelè et al., 2017) that each one of these inflow characteristics generates a specific response in the rotor at the 1P frequency. This is a very desirable feature, because:

- The 1P frequency is strongly dominated by these four "deterministic" characteristics of the wind, and much less so by turbulent fluctuations (Bertelè et al., 2017). On the other hand, higher response frequencies are associated with smaller-scale variations of the flow in space and time caused by turbulence (Bertelè et al., 2017).

- The measurement of such low frequencies requires low sampling rates, which eases the requirements on the sensors.

- There should be limited variability in such low frequencies among different installations of a same wind turbine type. Although this has not been demonstrated yet, it would imply that, once the method has been tuned on one machine, it should be applicable with minimum recalibration also on different turbines of the same type.

- The lower frequencies of the response of a wind turbine should be reasonably well captured by existing simulation tools used for design and certification. This implies that the method can be tested in a simulation environment, with the expectation of realistic results on its performance. This is clearly important for a number of reasons, not least the fact that in a simulation environment the actual inflow at the rotor disk is precisely known, something that is much harder to do in the field. Indeed, as shown later on, the incomplete knowledge of the actual inflow is one of the main limitations of the present experimental study, and of any field test.

If the turbine implements individual pitch control, the map correlating loads and wind characteristics is extended to include also blade pitch angle harmonics (Bertelè et al., 2018).

The load-harmonic method requires a training dataset consisting of measured rotor loads and corresponding measured wind characteristics. The dataset can be based on experimental measurements, or be generated synthetically using a simulation model; these two approaches were respectively termed model-free and model-based in Bottasso and Riboldi (2014). Here we consider the former approach. Indeed, a simulation model with the necessary characteristics might not always be available, for example in cases when wind sensing is applied to a turbine without the support of the manufacturer. Even when a model is available, it might not have been fully validated, so that a purely data-driven approach has a significant appeal.

It is envisioned that, in a practical application of the model-free harmonic-based method, the training phase would be a one-off activity performed at a test site equipped with a met-mast or other wind measuring devices such as lidars (Mikkelsen, 2014; Peña Diaz et al., 2014). Indeed, hub-tall met-masts are routinely used during certification (IEC, 2017), and could be employed for the additional purpose of training the observer. After training, the method could be used on other installations of that same turbine type at normal production sites without necessitating of met-masts or other devices. This claim, however, still remains to be demonstrated in practice.

The principal goal of this paper is to present the application of the load-harmonic estimator to field test data collected at a test site on a 3.5 MW wind turbine and a nearby met-mast (Schreiber et al., 2020; Bertelè and Bottasso, 2020). This experimental setup is a realistic representation of the scenario outlined above, where a hub-tall met-mast is located in close proximity of a wind turbine for certification purposes. From this point of view, the present dataset provides opportunities not only for a first —partial— field demonstration of the method, but also for addressing some important practical implementational aspects.

Specifically, the vertical shear requires special attention. In fact, a hub-tall met-mast with more than one anemometer can only measure the wind shear over the lower part of the rotor disk; on the other hand, the load-harmonic observer estimates a rotor-equivalent shear (i.e. a shear over the entire rotor disk area). For large modern rotors, half-rotor or full-rotor shears are not necessarily equal (Murphy et al., 2019; Schreiber et al., 2020). Therefore, a way is needed to extend the measurement of the inflow above the met-mast, possibly without resorting to extra wind-scanning equipment to reduce cost and complexity. This problem is solved here using the sector-effective wind sensing method (Bottasso et al., 2018; Schreiber et al., 2016, 2020). This second approach uses blade loads to estimate the average local speed over sectors of the rotor disk; from these sector-equivalent wind speeds, one can then estimate shears, including a vertical shear defined over just the lower half of the rotor.

The sector-effective speed and load-harmonic observers have distinct characteristics, which make them somewhat complementary and applicable to different scenarios. In fact, the sector-effective observer does not need to be trained with data before it can be used, since it is derived from standard performance characteristics of the rotor (Schreiber et al., 2020). Although not indispensable, field data can optionally be employed to fine-tune the observer, as shown in Schreiber et al. (2020). The sector-effective approach, however, can only reconstruct shears and not wind directions. The load-harmonic observer, on the other hand, can reconstruct both shears and directions but needs to be trained from data, which is a potential complication. Here, a novel three-step procedure is developed and demonstrated, where the two observers are used in synergy combining some of their complementary features:

1. The lower-half-rotor shear measured by the sector-equivalent speed method is tuned and validated with respect to the met-mast reference.

2. The full-rotor shear is computed using the validated sector-equivalent speed method, extending the measurement of the inflow above the met-mast.

3. This rotor-equivalent shear is finally used for training the harmonic-based estimator.

Although the present setup allows for a first demonstration of this procedure, it also presents some limitations that hinder a real and complete validation of the method. First, the extension of the shear above the met-mast is performed through the

same rotor loads that are also used by the harmonic-based estimator. Clearly, a completely independent measurement of the inflow up to the tip of the rotor would be preferable for validation purposes. Second, the present met-mast only includes a wind vane at hub height. This is a point-wise measurement, whereas the one provided by the observer —being obtained through the response of the rotor— is a rotor-effective quantity. Here again, it would be desirable to train and verify the method with an independently-derived rotor-equivalent quantity. Third, a met-mast cannot really provide a true and absolute ground truth, as it measures the flow away from the rotor disk (two and half diameters away, in the present case). When the wind is not directly aligned with turbine and mast, the wind shear and direction may be slightly different, on account of wind spatial variability, because of orographic and vegetation-induced effects. These differences are indeed visible to some extent in the present dataset. Even when wind, mast and turbine are aligned, the two measurements are not co-located and therefore not necessarily identical. Fourth, the met-mast does not provide measurements for two of the four observed quantities, namely horizontal shear and upflow, for which, consequently, no comparison nor conclusion can be made. Clearly, a more precise characterization of the effective inflow experienced by the rotor disk would be desirable for validation purposes. A lidar scanning the inflow immediately in front of the disk plane —to ensure co-location of the measurements— might be a possible solution.

Although the present study clearly falls short of a true validation of the harmonic-based formulation of wind sensing, it still provides for a first field demonstration of this method, giving also a useful insight into some of its main characteristics.

The paper is organized as follows. Section 2 describes the overall methodology, including a brief review of the harmonic-based estimator in §2.2 and a description of the test site and the measurement of the inflow characteristics in §2.3. The analysis of the wind observer performance is presented in Section 3, while Section 4 concludes the paper.

## 2 Methods

### 2.1 Wind parametrization

The wind inflow is described by four parameters: the vertical linear shear $\kappa_v$, the horizontal linear shear $\kappa_h$, the vertical wind misalignment angle (or upflow) $\chi$, and the horizontal (or yaw) misalignment angle $\phi$. These quantities are illustrated in Fig. 1 and are expressed in a hub-centered nacelle-attached reference frame, where $x$ is parallel to the axis of rotation (and it is therefore inclined with respect to the ground because of uptilt), $y$ is horizontal with respect to the ground and points left looking downstream, while $z$ forms a right handed triad. It should be noticed that the vertical shear is customarily defined with respect to the horizontal, instead of the uptilt, direction; additionally, its profile is typically either logarithmic or expressed as a power law, instead of a linear function. As explained later, these choices are made here to exploit the rotational symmetry of the rotor (Bertelè et al., 2019); this is useful in the present context, because it allows to overcome the lack of horizontal shear and upflow measurements in the available dataset. Clearly, the four wind parameters, once estimated, can be readily transformed into a horizontal frame, if necessary. Furthermore, abandoning the rotational symmetry, the observer can be formulated in terms of a vertical non-linear shear, as shown in Bertelè et al. (2017).

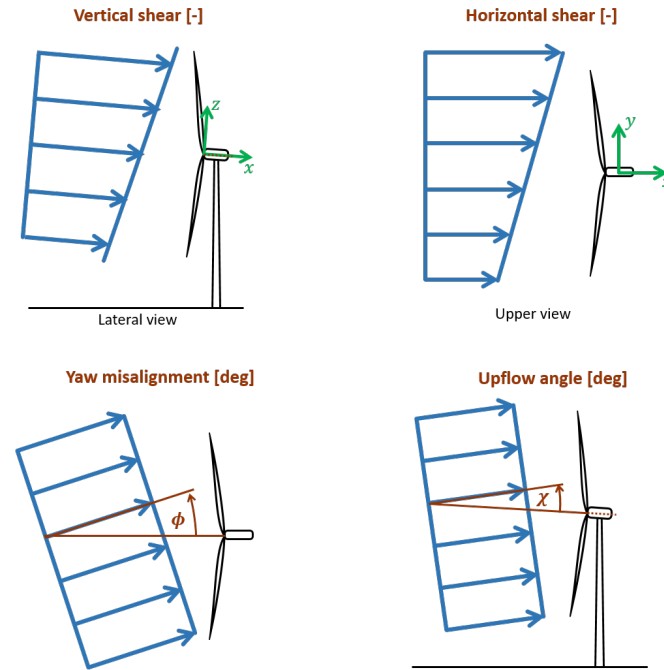

**Figure 1.** Definition of the four wind states used for parameterizing the wind field over the rotor disk.

A linearly sheared wind speed $W$ at the rotor disk is defined as

$$W(y,z) = V_h \left(1 + \frac{y}{R}\kappa_h + \frac{z}{R}\kappa_v\right), \tag{1}$$

where $V_h$ is the hub-height speed, and $R$ the rotor radius. By projecting the wind vector along the $x$, $y$ and $z$ axes, respectively, the three nacelle-attached velocity components $u$, $v$ and $w$ are readily obtained as

$$u(y,z) = W(y,z)\sqrt{1 - \tilde{v}^2 - \tilde{w}^2}, \tag{2a}$$

$$v(y,z) = W(y,z)\tilde{v}, \tag{2b}$$

$$w(y,z) = W(y,z)\tilde{w}, \tag{2c}$$

where $\tilde{v}$ and $\tilde{w}$ are defined as

$$\tilde{v} = \frac{v(0,0)}{V_h} = \sin\phi\cos\chi, \tag{3a}$$

$$\tilde{w} = \frac{w(0,0)}{V_h} = \sin\chi. \tag{3b}$$

For notational simplicity, the four wind parameters are grouped together in the wind state vector $\boldsymbol{\theta} = \{\tilde{v},\ \kappa_v,\ \tilde{w}, \kappa_h\}^T$. Given $\boldsymbol{\theta}$, the misalignment angles can be readily computed by inverting Eqs. (3) to get $\chi = \arcsin\tilde{w}$ and $\phi = \arcsin\tilde{v}/\cos\chi$.

## 2.2 Wind observer formulation

The relationship between wind states and rotor loads is assumed in the linear form

$$\boldsymbol{m} = \boldsymbol{F}(V,\rho)\boldsymbol{\theta} + \boldsymbol{m}_0(V,\rho) = [\boldsymbol{F}(V,\rho)\ \boldsymbol{m}_0(V,\rho)] \begin{bmatrix} \boldsymbol{\theta} \\ 1 \end{bmatrix} = \boldsymbol{T}(V,\rho)\,\overline{\boldsymbol{\theta}}, \tag{4}$$

where $\boldsymbol{F}$ and $\boldsymbol{m}_0$ are model coefficients. These coefficients depend on wind speed $V$ and air density $\rho$, on account of the different behavior, control and deformation of the machine in different operating conditions. For example, under the push of the rotor thrust, the tower will bend backward, in turn slightly changing the rotor uptilt; if this effect is not accounted for, this deformation-induced uptilt will affect the observed wind upflow. The dependency on wind speed is taken into account by discretizing the wind speed range in nodal values and linearly interpolating the model based on the rotor-equivalent wind speed (see §2.3.4), while density is accounted for as explained in §2.2.1. The load vector $\boldsymbol{m}$ is defined as

$$\boldsymbol{m} = \left\{ m_{1c}^{\mathrm{OP}},\, m_{1s}^{\mathrm{OP}},\, m_{1c}^{\mathrm{IP}},\, m_{1s}^{\mathrm{IP}} \right\}^T, \tag{5}$$

where $m$ indicates the blade bending moment, subscripts $(\cdot)_{1s}$ and $(\cdot)_{1c}$ respectively indicate 1P sine and cosine harmonic amplitudes, while superscripts $(\cdot)^{\mathrm{OP}}$ and $(\cdot)^{\mathrm{IP}}$ indicate out- and in-plane load components, respectively. Harmonic components are obtained from measured blade loads using the Coleman transformation (Coleman and Feingold, 1958), followed by low pass filtering.

The model coefficients $\boldsymbol{F}$ are not all independent, because of the rotational symmetry of the rotor (Bertelè et al., 2019). In fact, neglecting the disturbance caused by the tower, the effects on loads caused by a horizontal shear are the same as the ones caused by an equal vertical shear with a phase delay of $\pi/2$; the same holds true for the wind misalignment angles. This not only reduces the number of unknowns, but also eases the identification of the model, especially when using longer time averages. In fact, while both vertical and horizontal shear undergo rapid changes due to spatial turbulence variability, it is easier to observe slower changes in vertical shear than in the horizontal one. In fact, vertical shear exhibits slow natural changes over a significant range, for example because of diurnal fluctuations. On the other hand, horizontal shear might exhibit slow scale variability because of orographic effects or in waked conditions, which —depending on the turbine— might or might not happen very frequently or be particularly pronounced. Similarly, whereas yaw misalignment changes significantly in normal operation because of the inability of the yaw system to immediately and exactly track rapid wind direction fluctuations, upflow changes little (except that for orographic wind-direction-dependent effects). Therefore, by exploiting the rotational symmetry, a complete model can be identified simply from variable vertical shear and horizontal misalignment, because the effects of the other two wind states are obtained by the symmetry of the coefficients.

Dropping the dependency on $V$ and $\rho$ for notational simplicity, the model coefficients $\boldsymbol{T}$ are identified by stacking side by side measured wind states $\boldsymbol{\theta}$ into a matrix $\boldsymbol{\Theta} = \left[\overline{\boldsymbol{\theta}}_1, \ldots, \overline{\boldsymbol{\theta}}_N\right]$, while the corresponding measured blade loads $\boldsymbol{m}$ are stacked into matrix $\boldsymbol{M} = [\boldsymbol{m}_1, \ldots, \boldsymbol{m}_N]$, obtaining

$$\boldsymbol{M} = \boldsymbol{T}\boldsymbol{\Theta}. \tag{6}$$

The model coefficients are then computed by least squares as

$$\boldsymbol{T} = \boldsymbol{M}\boldsymbol{\Theta}^T \left( \boldsymbol{\Theta}\boldsymbol{\Theta}^T \right)^{-1}. \tag{7}$$

Measured loads $\boldsymbol{m}_{\mathrm{M}}$ are defined as

$$\boldsymbol{m}_{\mathrm{M}} = \boldsymbol{m} + \boldsymbol{r}, \tag{8}$$

where $\boldsymbol{m}$ is given by Eq. (4) and $\boldsymbol{r}$ is the residual with covariance $\boldsymbol{Q} = \mathrm{E}[\boldsymbol{r}\boldsymbol{r}^T]$. Residuals are assumed to be zero-mean and colored, and are due to measurement noise and unmodeled physics (Jategaonkar, 2015). Given the model coefficients, a maximum likelihood (Strutz, 2016) estimate $\boldsymbol{\theta}_{\mathrm{E}}$ of the wind states can be computed online during turbine operation from the measured loads $\boldsymbol{m}_{\mathrm{M}}$ from Eqs. (4) and (8) as follows

$$\boldsymbol{\theta}_{\mathrm{E}} = \left( \boldsymbol{F}^T \boldsymbol{Q}^{-1} \boldsymbol{F} \right)^{-1} \boldsymbol{F}^T \boldsymbol{Q}^{-1} (\boldsymbol{m}_{\mathrm{M}} - \boldsymbol{m}_0). \tag{9}$$

### 2.2.1 Density correction

Aerodynamic moments can be written as

$$m_{\mathrm{A}} = qARC(V, \rho), \tag{10}$$

where $q = 1/2\rho V^2$ is the dynamic pressure, $A = \pi R^2$ is the rotor disk area, while $C$ is a non-dimensional coefficient. A correction for density can be simply obtained as

$$m_{\mathrm{A_{ref}}} = m_{\mathrm{A_i}} \frac{\rho_{\mathrm{ref}}}{\rho_i}, \tag{11}$$

where $\rho_{\mathrm{ref}}$ is a reference density, and $\rho_i$ the density corresponding to measurement $m_{\mathrm{A_i}}$.

However, blade load sensors measure not only aerodynamic loads but also the effects of inertia and gravity, which do not depend on air density. Inertial loads for a balanced rotor spinning at constant rotor speed do not generate rotating 1P harmonic components, and hence do not appear in Eq. (4). On the other hand, gravitational terms generate 1P loads represented by the non-homogeneous term $\boldsymbol{m}_0$ in that same equation. According to Bertelè et al. (2017), this term can be written as

$$\boldsymbol{m}_0 = qARC(V, \rho) + \boldsymbol{g}. \tag{12}$$

The first term is a gravity-induced load due to the rotor deformation caused by aerodynamic loads; for example, if the blade bends under the push of thrust, the resulting deformation generates a non-null moment arm for gravity with respect to the blade root where the load sensor is located, resulting in a 1P load. This term is proportional to dynamic pressure and can be corrected for density. The second term $\boldsymbol{g}$ accounts for in-plane and out-of-plane gravity-induced loads, the latter being caused by blade precone, prebend and rotor uptilt. This term does not depend on density, and hence it should be eliminated by the equations before a density correction can be applied. To this end, first the model coefficients of Eq. (4) were identified for a very low wind speed, just above cut-in. Here the effects caused by $qARC$ are negligible, and hence $\boldsymbol{g} \approx \boldsymbol{m}_0$. Having first identified the gravity term $\boldsymbol{g}$ and then having eliminated it from model (4), each measured load was finally corrected for density using Eq. (11).

## 2.3 Wind parametrization in the field

Before wind states can be estimated at run time from measured loads using Eq. (9), the model coefficients must be identified through the simultaneous measurements of wind states and associated loads using Eq. (7). This section presents a practical method to perform this task, based on the use of a standard IEC-compliant (IEC, 2017) hub-tall met-mast. A similar procedure

could be used to identify the observer for a specific wind turbine type. Having obtained the model coefficients, one should be able to use the same observer for other installations of that same wind turbine type. Although there is yet no direct demonstration of this assertion, it seems reasonable to assume that wind turbines of the same model will have a similar 1P response to shears and misalignment angles. Additionally, Bottasso and Riboldi (2015) showed that the method is robust to the typical changes occurring in some of the wind turbine parameters across different installations of a same wind turbine type, including

changes in the stiffness of foundations, orographic effects, imbalance due to pitch misalignment, miscalibration of the load sensors and changes in airfoil lift and drag due to soiling/erosion.

### 2.3.1 Test site

Data was measured at a test site between October 19 and November 29, 2017, for a campaign unrelated to the present study. Since the data was collected long before the beginning of this work, the data had to be used "as is", without the possibility of

verifications, calibrations or any other activity meant at improving the knowledge of the conditions in which the dataset was collected.

    Figure 2 shows a panoramic view of the test site (Bromm et al., 2018), which is located in Germany a few kilometers inland from the Baltic Sea and is characterized by gentle hills, open fields and forests. A 3.5 MW eno114 turbine is installed at the site. The machine (labelled WT1 in the figure) has a 92 m hub height and a rotor diameter of 114.9 m.

A met-mast is situated at about 2.5 diameters (D) from the turbine. Wind direction was measured at a height above ground of 89.3 m with a Thies GmbH wind vane, while wind speed measurements were obtained with three cup anemometers produced from the same company and located at 89.3 m, 91.5 m and at the lower tip of the rotor (about 34 m). All measurements obtained on the mast were shifted in time on account of the distance between turbine and met-mast, the time delay being computed from the average wind speed.

A second turbine (labelled WT2) is also present on site, and its wake affects the met-mast and WT1 for easterly and southeasterly winds. Similarly, the wake of WT1 affects the met-mast for northern wind directions. All these conditions were discarded from the training dataset, in addition to all other situations when WT1 was not in a normal power production state. A forest of 15-20 m tall trees is located 300 m east of WT1; as only wind directions $\Gamma \in [180, 340]$ deg were considered in this work, this high roughness area was never in the inflow direction. On the other hand, the town of Brusow is located about

1 km to the west of the site, and its effects on the inflow are unknown. A test campaign conducted at the same site in the period July-November of the previous year revealed an almost equal distribution of unstable, neutral and stable conditions, as measured by an eddy covariance station (Bromm et al., 2018).

Synchronized turbine and blade load data was sampled at 10 Hz on WT1. Blades 1 and 3 were equipped with strain gages, installed in close proximity of the blade roots and measuring both flapwise and edgewise bending components; unfortunately, however, the same load sensors were not installed on blade 2. To reconstruct the missing load components, the measurements of blades 1 and 3 were shifted by $\pm 2\pi/3$, averaged together and then attributed to blade 2. This approximation assumes that neighboring blades experience the same loads when they are at the same azimuthal position, which is reasonable because loads and wind states are time-averaged quantities linked by a steady load-wind model (cf. Eq. (4)).

In general, sensors deployed in the field cannot be assumed to be always exactly calibrated, and they may suffer from a variety of issues that affect the quality of the measurements that they provide. To address this problem, it is useful to devise simple and practical ways to correct the measurements, even when the root cause of the problem is unknown. Here, consistent mismatches between the long-term mean readings of the two blade load sensors were observed. To correct for this inconsistency, the signals were adjusted a posteriori by a factor $s$, to enforce the same mean loads on the two blades. This was obtained by scaling the measurements as $\overline{m}_1(1+s) = \overline{m}_3(1-s)$, with $s = 0.0274$. Clearly, this is different from a true calibration meant to ensure the correct reading of a known quantity. However, since the data had been collected prior to this study and no additonal information was available, this is probably the only possible adjustment that can be applied. Additionally, the azimuth signal was corrected to account for sensor bias and dynamic effects, as explained in Schreiber et al. (2020). The yaw encoder signal was also corrected for an apparent inconsistency of its readings, as explained later in this section. The turbine on-board wind vane was found to correlate well with the signal provided by the mast, after correcting for time delays due to their different locations. However, for coherence with the reference wind speed measurements, also the wind direction reference was taken as the one provided by the mast.

## 2.3.2 Wind shears

The met-mast present at the test site reaches only up to hub height; this is also the typical case of IEC-compliant met-masts used for certification (IEC, 2017). The three anemometers at 34, 89 and 92 m can be used to estimate the vertical shear over the lower half of the rotor, which however in general differs from the shear computed over the whole rotor height.

To address this issue, the sector-effective wind speed (SEWS) estimation method described in Schreiber et al. (2020) was employed to obtain a rotor-effective reference for the shears. In short, the method works as follows: the blades are used as local speed sensors that, scanning the rotor disk, provide average speeds over four rotor quadrants. By using the two lateral and the lower quadrants, the shear over the lower part of the rotor disk can be computed. This quantity is validated with respect to the shear measured by the met-mast, assumed as a ground truth. Then, having verified a good correlation between the measured and estimated shears over the lower part of the rotor, the SEWSs for all four quadrants are used to calculate the wind shear over the whole rotor disk. A brief overview of the SEWS estimator is reported next, and the interested reader is referred to Schreiber et al. (2020) for further details.

The rotor cone coefficient is defined as

$$C_m\left(\beta, \lambda, q, \psi_i\right) = \frac{m_i}{0.5\rho ARV^2}, \tag{13}$$

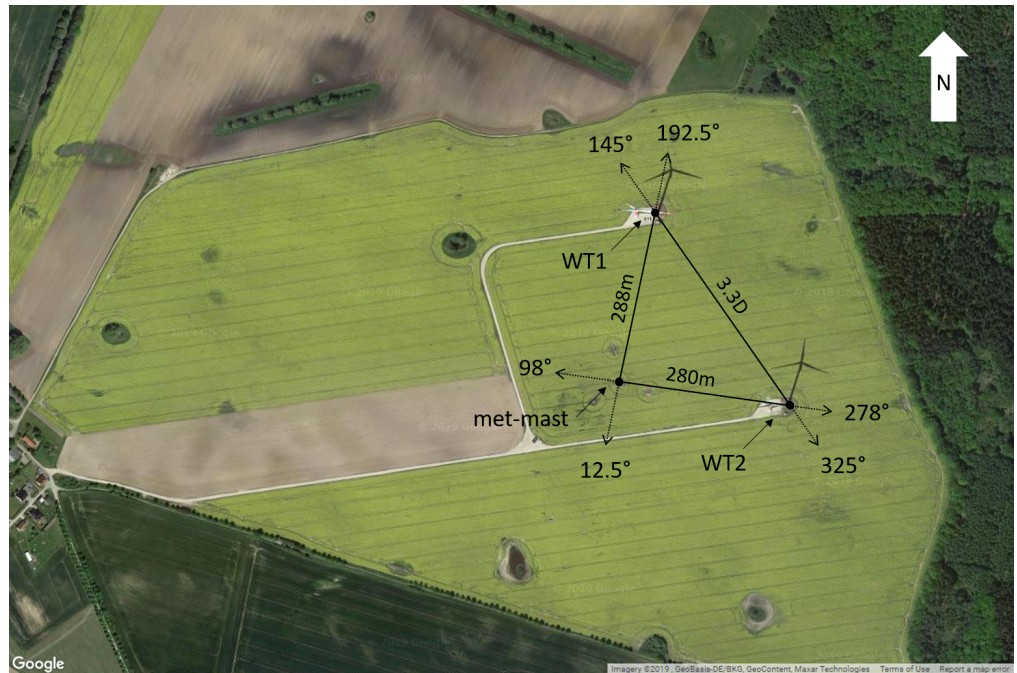

**Figure 2.** Satellite view of the test site, including waking directions and distances. WT1 indicates the turbine used for the present analysis (© Google Maps).

where $\beta$ is the pitch angle, $\lambda = \Omega R/V$ the tip speed ratio and $\Omega$ the rotor speed, $m_i$ the out-of-plane bending load of the $i$th blade and $\psi_i$ its azimuthal position. The dependency of the coefficient on the azimuthal position of the blade is primarily dictated by the effects of gravity, which for an uptilted rotor generate an out-of-plane bending moment that needs to be taken into account. Accuracy can be improved by considering the deformation of tower and rotor depending on operating condition (Bottasso et al., 2018), an effect that was neglected here for simplicity. Coefficient $C_m$ was computed from a complete aeroelastic model of the turbine, implemented with the code FAST (Jonkman and Jonkman, 2018). Inverting Eq. (13), a look-up table (LUT) is generated that returns the blade-effective wind speed $V_i$ given measured blade pitch angle, rotor speed, azimuthal blade position, bending moment and density:

$$V_i = \mathrm{LUT}_{C_m}\left(\beta, \Omega, \psi, m_i, \frac{\rho}{\rho_{\mathrm{ref}}}\right). \tag{14}$$

This way each individual blade is turned into a local wind speed sensor, which scans the rotor disk. Since this local measurement is noisy, the rotor disk is divided into sectors of area $A_{\mathrm{S}}$, and a sector-equivalent wind speed is computed as

$$V_{\mathrm{S}} = \int_{A_{\mathrm{S}}} V_i(\psi_i)\,\mathrm{d}A_{\mathrm{S}}. \tag{15}$$

Here the four sectors shown in Fig. 3 were used. This yields four measurements of the local speed at the rotor disk, namely above, below and to the sides of the hub center. Bottasso et al. (2018) showed that, for a linear shear and a 90-degree-wide sector, the SEWS corresponds to the inflow speed at a distance of approximately $2/3\,R$ from the hub center.

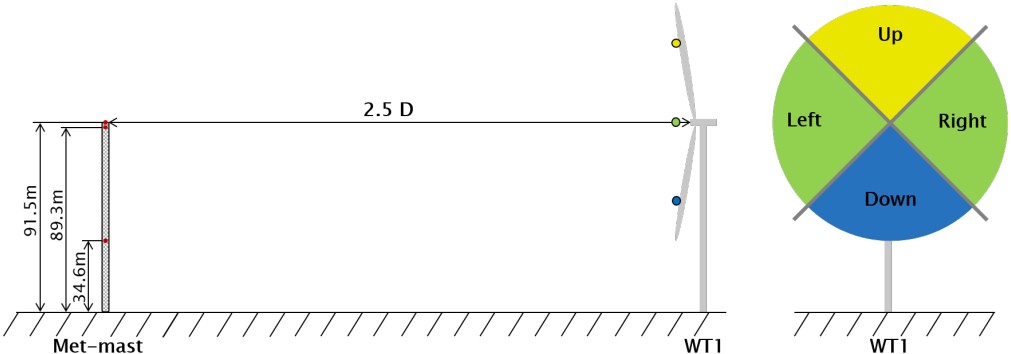

**Figure 3.** Definition of the four rotor sectors and their relative position with respect to the met-mast. Right: view looking downstream.

The rotor-effective horizontal linear shear can be computed inserting the SEWSs in Eq. (1) to get

$$\kappa_h = \frac{3}{2} \frac{V_{\mathrm{S,left}} - V_{\mathrm{S,right}}}{V_{\mathrm{S,left}} + V_{\mathrm{S,right}}}. \tag{16}$$

The analysis of the vertical shears requires some care. In fact, the linear vertical shear estimated by the met-mast and by the sector-effective speeds are computed from measurements obtained at different heights above ground; as such, they are not directly comparable, because shear has typically a non-linear variability with height. To address this issue, a power law is first fitted to the measurements to accurately represent the shape of the wind speed gradient; once the power law parameters have been determined, linear shears are computed for mast and observer between the same two heights, resulting in comparable quantities. As previously mentioned, the vertical shear can be parameterized in various ways. In this work, a linear fit was chosen in order to match the linear definition of the horizontal shear, because this avoids the need for horizontal shear measurements by using the rotor symmetry (Bertelè et al., 2019).

More precisely, the calculation of the linear shears is conducted as follows. The power law profile is defined as

$$V_{\mathrm{PL}}(z) = V_{\mathrm{ref}} \left( \frac{z + H}{H} \right)^{\alpha}, \tag{17}$$

where $H$ is the height of the hub, $V_{\mathrm{ref}}$ the wind speed at that point, and $\alpha$ the power law exponent. Given $n$ measurements $V_i$ at $z_i$, the parameters of the power law are computed by the following best fit:

$$(V_{\mathrm{ref}}, \alpha) = \arg \min_{V_{\mathrm{ref}}, \alpha} \sum_{i=1}^{n} \left( V_{\mathrm{PL}}(z_i) - V_i \right)^2. \tag{18}$$

Notice that two measurements at two different heights are sufficient to estimate the power law, since it depends on only the two free parameters $V_{\mathrm{ref}}$ and $\alpha$. Having solved the fitting problem (18), the linear shear $\kappa_v$ between two generic heights $z_A$ and $z_B$

is computed as

$$\kappa_v = \frac{R\left(V_{\text{PL}}(z_A) - V_{\text{PL}}(z_B)\right)}{z_A V_{\text{PL}}(z_B) - z_B V_{\text{PL}}(z_A)}. \tag{19}$$

The left plot of Fig. 4 shows the correlation between 10-min averages of the vertical shears obtained by the met-mast and by the sector-effective wind speeds on the lower half of the rotor. Only wind directions between 170 and 215 deg are considered, where the turbine and met-mast are aligned. The power law for the met-mast was obtained by using all three speed measurements, although the two at 89.3 and 91.5 m above ground are almost coincident. For the sector-effective observer the power law was obtained by using the two estimates $(V_{\text{S,left}} + V_{\text{S,right}})/2$ at $z = 0$, and $V_{\text{S,down}}$ at $z = -2/3\,R$ (although this latter value is strictly valid only for linear shears). For both cases, the power law coefficients were first computed using Eq. (18), and then the lower-half-rotor linear shear was obtained from Eq. (19) using $z_A = 0$ and $z_B = -R$. The figure shows that there is a good correlation between the two lower-half-rotor shears, resulting in a Pearson's coefficient of 0.91.

The figure also shows that the linear fit (red dashed line) has a different slope than the ideal match (black solid line). This could be due to a non-ideal power law profile, but also by a non-exact elimination of the effects of gravity, for example because of a different position of the load sensors in the model and reality or a slightly modified uptilt on account of tower deformation. Unfortunately, not enough information on the present experimental setup was available to determine the cause of this discrepancy with certainty. However, the results presented later in Section 3 were pragmatically corrected to account for this error: the slope deviation was evaluated from Fig. 4, and the estimates were modified accordingly to yield corrected results lying on the bisector.

For the same data points, the right plot of Fig. 4 shows the correlation between the vertical shears obtained by the met-mast and by the sector-effective estimator over the complete rotor. Here again the power law for the met-mast was obtained by using all three speed measurements. For the sector-effective estimator the power law was obtained by using Eq. (18) with the three estimates $V_{\text{S,up}}$ at $z = 2/3\,R$, $(V_{\text{S,left}} + V_{\text{S,right}})/2$ at $z = 0$, and $V_{\text{S,down}}$ at $z = -2/3\,R$, although here again the vertical coordinates are strictly valid only for a linear shear. For both cases, the full-rotor linear shear was computed from Eq. (19) using $z_A = R$ and $z_B = -R$. It should be noted that, since the height of the top anemometer reaches only up to hub height, for the met-mast the calculation of the full-rotor shear implies a considerable extrapolation outside of the available measurements.

Comparison of the right and left plots of Fig. 4 shows that, in the full-rotor case, there is a lower correlation between the met-mast and the SEWS observer than in the lower-half rotor case. This indicates that the shear changes over the height of the rotor disk. In addition, as expected for a typical power law where the profile gradient increases with height, the lower-half-shear coefficient is typically higher than the full-rotor one.

Based on these results, it appears that the rotor-effective shear used for identifying the model of §2.2 would require a tall met-mast or other wind measurement devices such as lidars or sodars capable of scanning the inflow reaching the top of the rotor. Here —as such a tall mast was not available— an alternative approach was adopted: the sector-equivalent wind speed method was used to virtually extend the met-mast measurements to the required height. Based on the good correlation shown by the left plot of Fig. 4 for the lower-half-rotor shear, it was concluded that the two lateral and the lower sector-equivalent speeds are sufficiently accurate for the purpose of estimating shears. Since the top sector speed is based on exactly the same

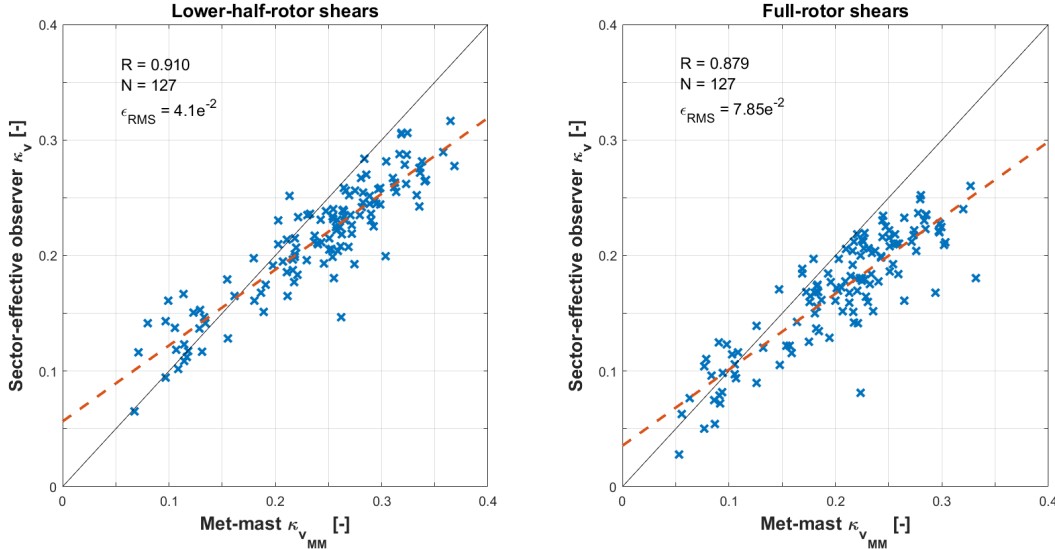

**Figure 4.** Correlation between 10-min averages of the vertical linear shears measured with the met-mast and the sector-effective observer. Left: lower-half rotor shears; right: full-rotor shears. Red dashed line: linear best fit; black dashed line: ideal match; $R$: Pearson's correlation coefficient; $N$: number of data points; $\epsilon_{\mathrm{RMS}}$: root mean square error.

calculation procedure as the other ones, all four speeds were then used to estimate the full-rotor shear, which in turn was adopted as reference for the identification of the model of §2.2.

Unfortunately a similar validation cannot be performed for the horizontal shear with the present met-mast, because of the lack of multiple lateral measurements. However, the horizontal shear is based on the same sector-equivalent wind speeds that

estimate the vertical shear with good accuracy, so that there is no reason to believe that Eq. (16) should not provide a similarly good-quality estimate. Additionally, the horizontal shear based on the two lateral sector-effective wind speeds was shown in Schreiber et al. (2020) to track the movement of an impinging wake with remarkable accuracy.

### 2.3.3   Wind misalignment angles

The met-mast is equipped with a single wind vane measuring the wind direction $\Gamma$ at hub height. Unfortunately, this means that

only a point-wise measurement is available, instead of the rotor-equivalent one that would be ideally necessary for the training of the load-harmonic method of §2.2. This is a limit of the current setup and of the present attempt at validating the approach. Nonetheless, a pragmatic choice was made here to use this signal as a proxy for the rotor-effective horizontal wind direction. The misalignment angle between turbine and wind was obtained by subtracting the absolute yaw angle of the nacelle from the met-mast-measured wind direction. The result was filtered with a 1-min moving average to remove the faster fluctuations.

The top plot of Fig. 5 shows 10-min averages of the resulting met-mast yaw misalignment angle $\Phi_{\mathrm{MM}}$, plotted as a function of wind direction $\Gamma$. The clear trend visible in the plot is probably due to a miscalibration of the nacelle yaw encoder. Indeed,

Bromm et al. (2018) also noticed a non-constant offset when comparing the turbine SCADA orientation with the one provided by a temporarily installed GPS system. This trend was removed using the first ten days of data, excluding waked directions, obtaining the bottom plot of Fig. 5.

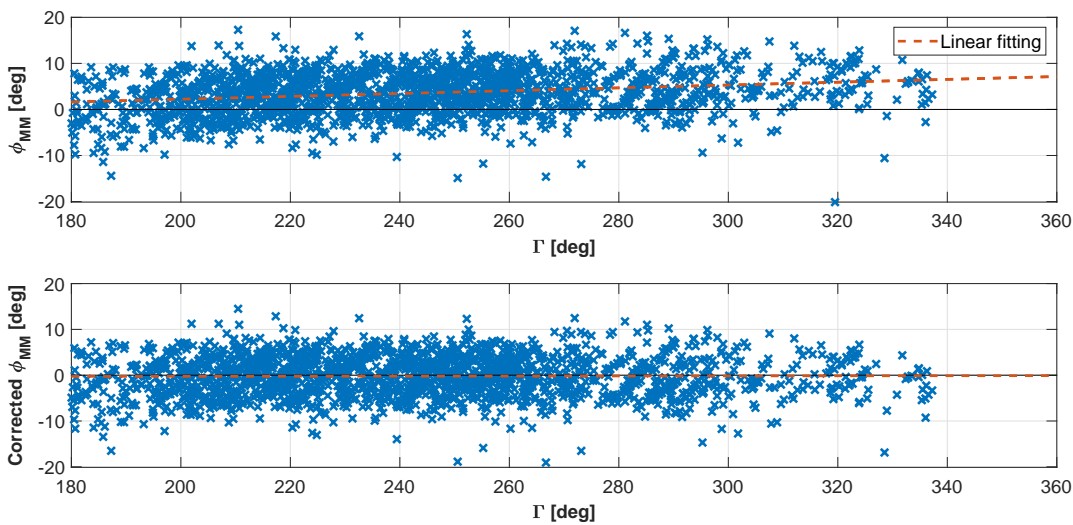

**Figure 5.** 10-min averages of met-mast horizontal wind misalignment angle $\phi_{\mathrm{MM}}$ vs. wind direction at the met-mast $\Gamma$, before (top) and after (bottom) correction for yaw encoder error.

As the current setup does not provide for measurements of the upflow, the rotational symmetry of the rotor was used to compute the relevant model coefficients.

### 2.3.4 Wind speed and density

Since the load-wind model expressed by Eq. (4) depends on the operating conditions, a rotor-effective wind speed was computed with the torque balance equation (Ma et al., 1995; Van der Hooft and Engelen, 2004; Soltani et al., 2013; Schreiber et al., 2020) and used as scheduling parameter of the wind observer. Figure 6 shows an excellent correlation for the 10-min averages of the computed rotor-effective wind speed and the met-mast hub-height speed, with a Pearson coefficient of 0.988 and a root mean square (RMS) error $\epsilon_{\mathrm{RMS}} = 0.418$ ms$^{-1}$. Density was obtained from the ideal gas law based on temperature, since no additional information was available, and was used to rescale the load measurements.

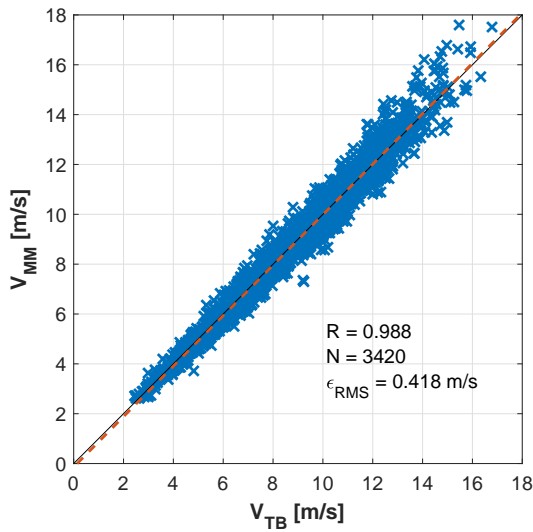

**Figure 6.** Correlation between 10-min averages of met-mast hub-height wind speed $V_{\mathrm{MM}}$ and rotor-effective wind speed $V_{\mathrm{TB}}$ estimated with the torque balance equation. Red dashed line: linear best fit; black dashed line: ideal match; $R$: Pearson's correlation coefficient; $N$: number of data points available; $\epsilon_{\mathrm{RMS}}$: root mean square error.

## 3 Results

### 3.1 Model identification

The observer coefficients were identified with Eq. (7) using the horizontal and vertical shears obtained from the sector-effective wind speeds, and the yaw misalignment angle computed from the met-mast wind vane and the nacelle yaw encoder, corrected according to Fig. 5. The upflow model coefficients were obtained from the rotational symmetry of the rotor behavior. Load measurements were corrected for density, the reference value being set to $1.238 \, \mathrm{kg m^{-3}}$.

The model coefficients were scheduled as functions of the rotor-effective wind speed computed from the torque balance equation. The wind speed nodes of the linear parameter varying (LPV) model (4) were defined as $V = [4, 5, 6.5, 8, 9, 10, 12, 13.5] \, \mathrm{ms^{-1}}$. This means that model coefficients were computed at each of these wind speed nodes, while any speed within the range $[4, 13.5] \, \mathrm{ms^{-1}}$ —i.e. not necessarily at the nodes— was used for identification, by linearly distributing its contributions to the two neighboring nodes. At run time, the coefficients were interpolated from the LPV based on the current wind speed.

Table 1 shows the range covered by each parameter within the training dataset.

About 15% of the available data was used for identification, leaving about 370 hours of measurements for validation. In the following, the performance of the harmonic observer is evaluated solely based on the validation dataset, i.e. excluding all data points used for training.

**Table 1.** Minimum and maximum values of rotor effective wind speed, turbulence intensity (TI), density, yaw misalignment, vertical and horizontal shear within the training dataset.

|  | $V$ [ms$^{-1}$] | TI [%] | $\rho$ [kgm$^{-3}$] | $\phi_{\mathrm{MM}}$ [deg] | $\kappa_v$ [-] | $\kappa_h$ [-] |
|---|---|---|---|---|---|---|
| min | 3.89 | 1.15 | 1.221 | $-12.66$ | $-0.045$ | $-0.053$ |
| max | 13.68 | 11.06 | 1.256 | 8.28 | 0.242 | 0.087 |

### 3.2 Wind observer performance

Models were identified based on different time averages of the raw 10 Hz data. Here, the two cases of 1-min and 10-min averages are presented, because they correspond to the typical outputs of standard SCADA systems. In both cases, the raw data points were the same; this means that the 1-min model was identified on 10 times more load-state pairs than in the case of the 10-min model.

An overview of the performance of the two models is given by Fig. 7 (for the 10-min case) and 8 (for the 1-min case). The figures report correlations between reference and observed parameters, using the validation sub-set for wind speeds above 8 ms$^{-1}$. For each parameter, one per subplot, the reference state is shown on the $x$ axis, whereas the observed one on the $y$ axis.

Comparison of the 10-min and 1-min cases shows that results are essentially identical for the shears. For the misalignment angle, results are very slightly better using the longer time window, notwithstanding the smaller number of load-state pairs used for identification. Probably this is because longer time averaging alleviates the effects of outliers. Based on these results, the rest of the paper only considers the 10-min case.

Considering the shears, Fig. 7 shows that the Pearson's correlation coefficients ($R$) is above 0.9, and the RMS error $\epsilon_{\mathrm{RMS}}$ is of the order of $10^{-3}$. The yaw misalignment angle is less accurate, possibly because the reference is point-wise whereas the estimate is rotor-effective. Indeed, investigations at the same site with a more complete setup including a lidar profiler reported significant veer at the inflow (Bromm et al., 2018). However, with a correlation coefficient of 0.85 and an $\epsilon_{\mathrm{RMS}}$ of 1.9 deg, the matching is still good.

It is very interesting to observe that, even a model trained only with 10-min averages, is still able to provide for time-resolved estimates of the parameters. To illustrate this fact, Fig. 9 reports a 10 Hz time history of the vertical shears from the validation sub-set. The figure corresponds to about two days of operation, during which the wind direction (bottom plot) was $\Gamma \in [145, 260]$ deg. Turbine and met-mast are roughly aligned for $\Gamma \in [177.5, 215]$ deg; WT1 is in the wake of WT2 for approximatively $\Gamma \in [120, 170]$ deg, the two directions being indicated in the plot with two horizontal dashed lines. The top plot of the figure shows the lower-half-rotor shears measured at the met-mast and by the sector-equivalent speeds. Although some discrepancies are present, the figure shows that the sector-effective observer is capable of following the main changes in shear captured by the met-mast. The main discrepancies can be found between 2PM of October 21 and about 4AM of October 22, when WT1 is in the wake of WT2 or in its close proximity. However, one should not forget that the two estimates correspond to two locations spaced 2.5D apart, and that the exact ground truth at the rotor disk —where the observers operate— is unknown.

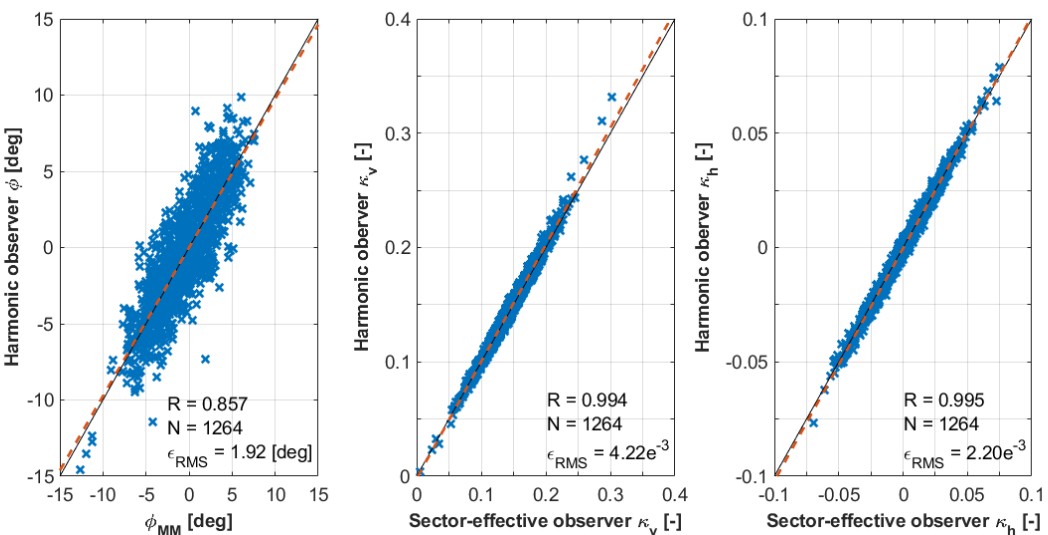

**Figure 7.** Correlation of 10-min averages between estimated parameters ($y$ axis) and their reference quantities ($x$ axis) for $V \geq 8 \text{ ms}^{-1}$. From left to right: yaw misalignment angle, vertical linear shear, horizontal linear shear. Red dashed line: linear best fit; black dashed line: ideal match; $R$: Pearson's correlation coefficient; $N$: number of data points; $\epsilon_{\text{RMS}}$: root mean square error.

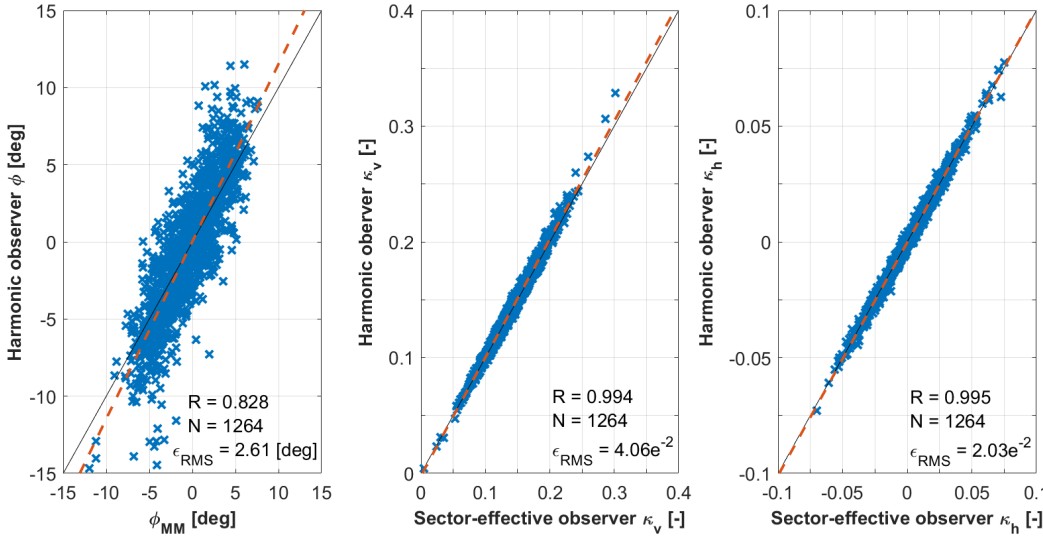

**Figure 8.** Correlation of 1-min averages between estimated parameters ($y$ axis) and their reference quantities ($x$ axis) for $V \geq 8 \text{ ms}^{-1}$. From left to right: yaw misalignment angle, vertical linear shear, horizontal linear shear. Red dashed line: linear best fit; black dashed line: ideal match; $R$: Pearson's correlation coefficient; $N$: number of data points; $\epsilon_{\text{RMS}}$: root mean square error.

The central plot of the same figure shows the rotor-equivalent shear estimated by Eq. (9) based on rotor harmonics and its reference quantity obtained by the sector-equivalent speeds. The two vertical shears are in excellent agreement, even with respect to relatively fast fluctuations.

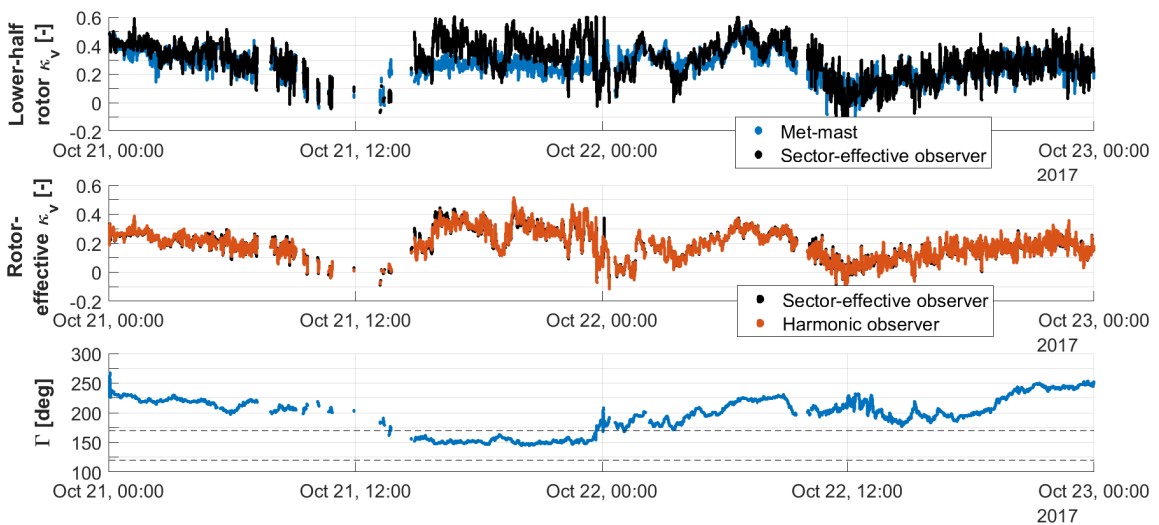

**Figure 9.** Time history of vertical shears at 10 Hz. From top to bottom: lower-half-rotor shear from the met-mast (blue) and the sector-effective observer (black); full-rotor-equivalent shear using Eq. (9) (red) and reference from the sector-effective observer (black); wind direction measured at the met-mast, with WT1 in the wake of WT2 between 120 and 170 deg (dashed horizontal lines).

To provide for a more complete statistical characterization of the observer performance, the 10-min data points were binned for the various relevant parameters. For each bin, the mean absolute error (MAE) between the estimated $\theta_\mathrm{E}$ and reference $\theta_\mathrm{R}$ wind parameter was computed as $\epsilon = 1/N \sum_i^N |\boldsymbol{\theta}_{\mathrm{R}_i} - \boldsymbol{\theta}_{\mathrm{E}_i}|$.

Figure 10 shows the MAE $\epsilon$ for yaw misalignment (top left), vertical and horizontal shear (top and bottom right, respectively), plotted as functions of binned wind speed, for various binned turbulence intensity (TI) levels. The number of available hours of data is reported in the bottom left histogram of the figure, to help determine the statistical significance of the results. Looking at the yaw angle results, it appears that the maximum error is about 3 deg and that accuracy tends to increase for higher wind speeds. Moreover, TI appears to play only a small effect on the results. As previously mentioned, this can be attributed to the fact that 1P harmonics are dominated by the for wind states, and only modestly affected by turbulent fluctuation.

The error in the vertical shear includes the error between the met-mast and the sector-effective observer of §2.3.2. Even in this case the error is small, and effects of TI are present but relatively mild. The figure also reports the horizontal shear, whose error —although very small— might not be very indicative: since no reference value was available from the met-mast for this quantity, only the error with respect the to sector-effective observer of §2.3.2 could be quantified.

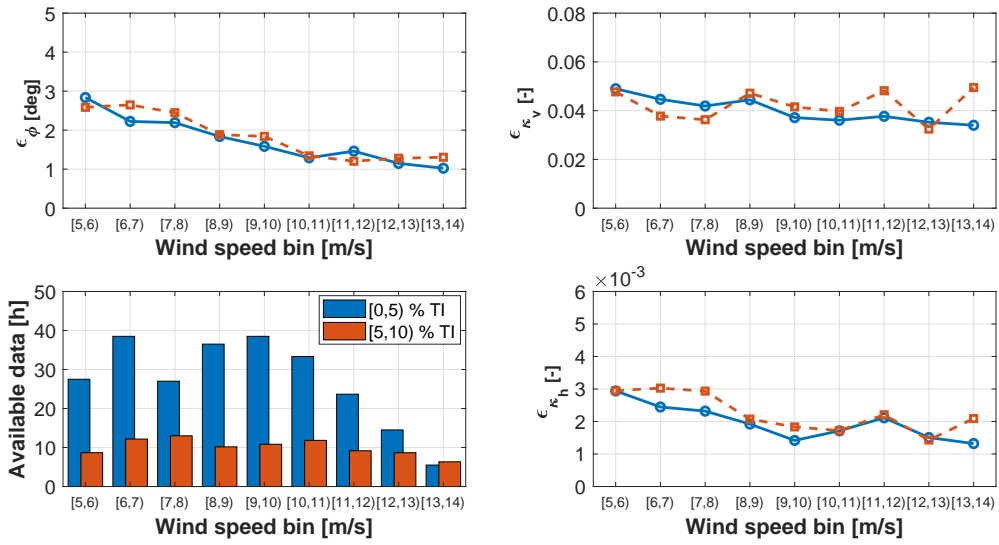

**Figure 10.** MAE $\epsilon$ vs. binned rotor-effective wind speed, for binned TI. Top left: yaw misalignment; top right: vertical shear; bottom right: horizontal shear; bottom left: hours of available data.

Figure 11 reports the results for varying binned air density. The plots show that the density correction of §2.2.1 is not perfect, probably because of an only approximate identification of the gravity term in Eq. (12).

Finally, Fig. 12 reports the results for varying wind direction. Looking at the vertical shear, the best results are obtained for wind directions between 170 and 210 deg, when turbine and met-mast are aligned, whereas the error increases significantly for other wind directions. When turbine and met-mast are not aligned, the two can be subjected to slightly different inflows, on account of orographic and vegetation-induced effects. This indicates once again that, as noted earlier on, the information provided by the reference met-mast cannot be regarded as an absolute ground truth. The yaw misalignment angle seems to be less influenced by these local effects, which might induce stronger local changes in shear than in direction at this particular site.

## 4 Conclusions

This paper has presented the application of a previously published harmonic-based wind sensing method to an experimental dataset. The setup at the test site is not complete enough to provide for a true field validation of the method. However, it is representative of a practical scenario where, by using a hub-tall certification met-mast, the method is trained for a given turbine model, before being deployed on assets of that same type at other production sites. After having explained the methodology and described the test site, the paper has also formulated a new method to extend the shear measured by a hub-tall mast to the tip of the rotor, in order to compute a full-rotor shear.

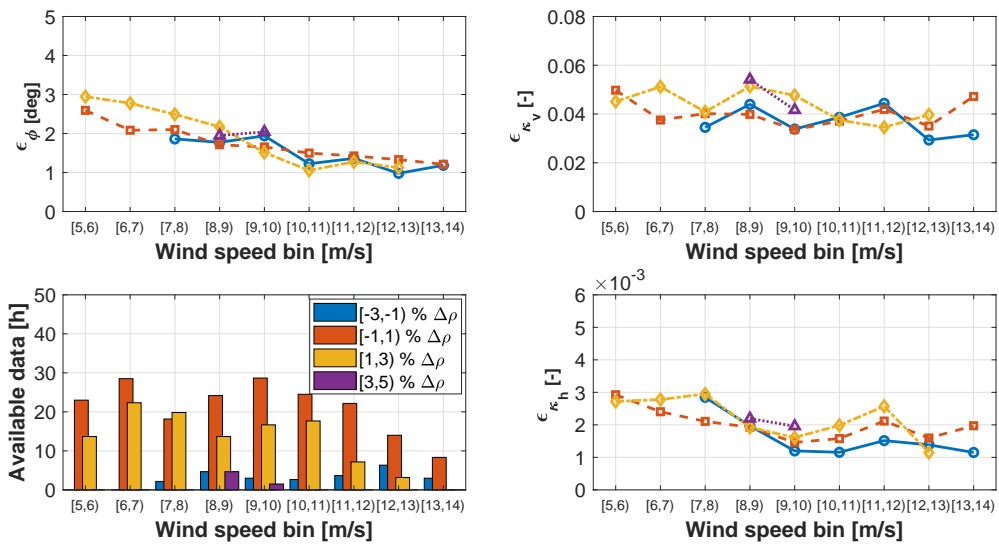

**Figure 11.** MAE $\epsilon$ vs. binned rotor-effective wind speed, for binned density change $\Delta\rho$ wrt. standard air. Top left: yaw misalignment; top right: vertical shear; bottom right: horizontal shear; bottom left: hours of available data.

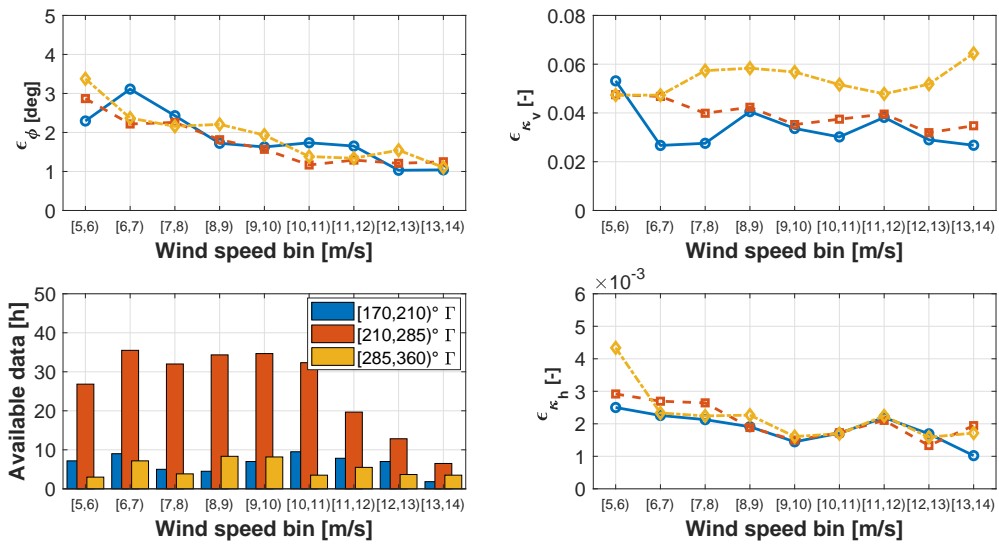

**Figure 12.** MAE $\epsilon$ vs. binned rotor-effective wind speed, for binned wind direction $\Gamma$. Top left: yaw misalignment; top right: vertical shear; bottom right: horizontal shear; bottom left: hours of available data.

Based on the results analyzed herein, and notwithstanding the limits of the present dataset, the following conclusions can be drawn:

– There is a good correlation between met-mast and estimated lower-half rotor shears, with Pearson's coefficients above 0.9 and RMS errors around $4\mathrm{e}^{-2}$;

– There is an excellent correlation between the full-rotor shear extended above the mast and the one estimated by harmonic loads, with Pearson's coefficients above 0.99 and RMS errors around $4\mathrm{e}^{-3}$;

– Training with 1-min or 10-min averages produces shear estimates of a very similar quality, but there is a marginal improvement of the wind direction for the longer time window. This is probably due to the noisier nature of wind direction, which is measured here only at hub height.

– Notwithstanding a training based on 10-min averages, the quality of the correlation between estimates and references does not only apply to 10-min quantities, but it also extends to time-resolved 10 Hz signals. In this sense, the observer seems capable of following relatively fast changes in shear. This might be useful for certain application scenarios, as for example the tracking of horizontal shears induced by wake interactions.

– There is a non-negligible effect of non-exact wind-mast-turbine alignment. In this sense, the actual quality of the correlation might be even better than what appears from the results shown here. This is in fact an intrinsic limit of field testing, where an exact ground truth is in general difficult if not impossible to obtain. Realistic simulations and wind tunnel studies as the ones reported in Bertelè et al. (2017, 2018, 2019) —where the ground truth is known— may help in this sense.

– Yaw misalignment is also estimated with reasonable quality, maximum errors being in general below 3 deg. However, the results here are less conclusive due to the fact that the met-mast reference is a point-wise measurement that might not fully represent rotor-effective conditions.

– There is only a modest effect of TI, which supports the hypothesis that 1P harmonics are mostly driven by "deterministic" wind characteristics and less affected by turbulent fluctuations.

– Notwithstanding the complicated effect of gravity on harmonic load components, its presence can be eliminated with enough accuracy to allow for a reasonably precise density correction.

The main limits of the present dataset are as follows: independent reference measurements for horizontal shear and upflow were completely missing, yaw misalignment was measured only at a point instead of over the rotor disk, and the vertical shear had to be extended over the hub by the use of another estimation method. Although the utmost care was put into the reconstruction of the full-rotor vertical shear, this operation still had to rely on the same blade load measurements used by the harmonic estimator, which is clearly a weakness. Additionally, as the test campaign was performed prior to the present study,

the dataset had to be used as is, without any possible verification, correction or calibration of the sensors. Other less substantial limitations are also present, for example caused by the missing load sensors on one of the blades.

A continuation of this work would greatly benefit from access to a more complete dataset. Multiple, independent rotor-effective measurements of the inflow in close proximity of the rotor disk would be necessary to establish an effective ground truth. This would enable a better characterization of the accuracy of this method, and to study the effects induced by training with a standard hub-tall mast. A remaining open point is the sensitivity of the method to phenomena like aging, soiling and rotor imbalances. Indeed, any exogenous cause affecting the 1P response will be interpreted by the harmonic-observer as a change in the wind states. Some reassuring results have already been reported by Bottasso and Riboldi (2015), although a more thorough experimental investigation is necessary. Finally, it remains to be shown that the method can indeed be trained on a turbine and, then, applied to another machine of that same model at another site; although this seems to be a very reasonable assumption, the evidence that this is indeed possible is lacking.

*Acknowledgements.* The authors express their gratitude to Stefan Bockholt and Alexander Gerds of eno energy systems GmbH, who granted access to the measurement data and turbine model, and to Marijn van Dooren, Anantha Sekar and Martin Kühn of ForWind Oldenburg, who shared insight on the data. This work has been supported by the CompactWind II project (FKZ: 0325492G), which receives funding from the German Federal Ministry for Economic Affairs and Energy (BMWi).

## Nomenclature

| | |
|---|---|
| $A$ | Rotor area |
| $C_m$ | Cone coefficient |
| $H$ | Height of the hub above ground |
| $m$ | Blade bending moment |
| $\boldsymbol{m}$ | Vector of moment harmonics |
| $N$ | Number of available data points |
| $q$ | Dynamic pressure |
| $R$ | Rotor radius or Pearson's coefficient |
| $\boldsymbol{Q}$ | Covariance matrix |
| $V$ | Wind speed |
| $V_h$ | Wind speed at hub height |
| $V_{\mathrm{PL}}(z)$ | Power law wind speed profile |
| $V_S$ | Sector-effective wind speed |
| $V_{\mathrm{TB}}$ | Torque-balance rotor-effective wind speed |
| $\tilde{v}$ | Non-dimensional tangential cross-flow at hub height |
| $\tilde{w}$ | Non-dimensional vertical cross-flow at hub height |

| | | |
|---|---|---|
| | $x, y, z$ | Hub-centered nacelle-attached axes |
| | $\beta$ | Pitch angle |
| | $\Gamma$ | Wind direction |
| | $\epsilon$ | Mean absolute error |
| 5 | $\boldsymbol{\theta}$ | Wind state vector |
| | $\kappa_h$ | Horizontal shear |
| | $\kappa_v$ | Vertical shear |
| | $\lambda$ | Tip speed ratio |
| | $\rho$ | Air density |
| 10 | $\phi$ | Yaw misalignment angle |
| | $\chi$ | Upflow angle |
| | $\psi$ | Azimuth angle |
| | $\Omega$ | Rotor speed |
| | $(\cdot)^T$ | Transpose |
| 15 | $(\cdot)^{\mathrm{IP}}$ | In-plane component |
| | $(\cdot)^{\mathrm{OP}}$ | Out-of-plane component |
| | $(\cdot)_{1c}$ | 1P cosine amplitude |
| | $(\cdot)_{1s}$ | 1P sine amplitude |
| | $(\cdot)_{\mathrm{E}}$ | Estimated quantity |
| 20 | $(\cdot)_{\mathrm{MM}}$ | Met-mast measurement |
| | $(\cdot)_{\mathrm{ref}}$ | Reference quantity |
| | $(\cdot)_{\mathrm{RMS}}$ | Root mean square |
| | 1P | Once per revolution |
| | MAE | Mean absolute error |
| 25 | Lidar | Light detection and ranging |
| | LUT | Look-up table |
| | RMS | Root mean square |
| | SEWS | Sector-effective wind speed |
| | Sodar | Sound detection and ranging |
| 30 | TI | Turbulence intensity |
| | WT | Wind turbine |

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
