# Peer review of "Wind inflow observation from load harmonics: initial steps towards a field validation"

_Wind Energy Science, 2020_

## Referee Comment (RC1) · Anonymous Referee #1 · 19 Aug 2020

A good account of very careful, detailed work on a technique which could have useful applications, with a good attempt to account for the inevitable difficulties of a real field test.

Detailed comments (P=page, L=line):

P2 L4: the term "lower spectrum" is not clear - or is it a typo?

P3: in L2 "does not need to be trained with data" but in L7 "is tuned" - what sort of tuning (and isn't that a simple form of training with data)?

P4 L8: change "write" to "can be written as" or "are given by"

P4 Figure 1: top left figure (vertical shear) shows arrows originating from a non-vertical

line. Does this represent rotor tilt? It should be explicitly stated somewhere that rotational symmetry depends on the coordinate system being tilted to align with the shaft axis. Presumably, after estimation you would need a final step to transform the wind components back to the 'normal' (untilted) reference frame.

P5 L21: "horizontal shear does not (except in waked conditions)" - probably not very much on average, I agree, but still maybe a bit sometimes, at least onshore due to orographic / vegetation effects - but anyway there can be significant, though short-lived, stochastic changes in horizontal (and vertical) shear across the rotor due to spatial turbulence effects.

P6 L4: I think you should explain how Equation (8) is derived, and where Q comes from.

P7 L5: "fairly robust to changes" - maybe "fairly robust to typical changes"? Presumably can't be true in the case of large changes (hopefully unlikely).

P7 L27: Using phase-shifted measurements from blade 1 & 3 to estimate the load on blade 2: Presumably this actually means time-shifted using rotor speed? Probably needs an equation here, and a some justification for the absence of a blade 2 measurement not affecting the results. Can you say why there wasn't a measurement on blade 2?

P9 L12: presumably the reference explains in more detail, but can you give a justification for using 2/3 R? Wouldn't it depend on the turbine aerodynamic details?

P10 equations 16 & 17: Consistency of notation: is VPL(z) the same as V(z)PL?

P10 L10: why was a 50-degree sector chosen, and why is it not exactly centred on the direction separating mast and turbine?

P10 L13: "two measurements" - presumably these are actually estimates rather than measurements?

P10 L14: Since the shear is actually non-linear, and different parts of the blade contribute differently to loading, some sort of weighted mean shear gradient might be more appropriate than the slope between hub and blade tip?

P10 L16: different slopes in Figure 4: Might this be due to non-linearity of shear, and/or the use of 2/3 R, or do you have some other explanation?

P11 L14: What filter characteristic, and how was it chosen? Are you filtering the wind vane signal, or the wind direction obtained by combining the wind vane signal with the nacelle position signal?

P12 L8/9: Are you effectively assuming zero upflow, as you can't measure it?

P13 L6: "scheduled as functions of the rotor-effective wind speed" - Not clear how you did this - was it by binning results in wind speed bins according to 10-minute average rotor-effective wind speed estimate and fitting model parameters for each wind speed bin?

Conclusions: "Training with 10-min data improves the quality of the estimates" stated without providing evidence. Can this be substantiated a bit better?

General comments:

The importance of veer is becoming more apparent especially for large turbines and stable conditions. How easily could the model be extended to provide an estimate of veer?

Some comment on how the model should be adjusted in case of a turbine which is using individual pitch control?

---

## Referee Comment (RC2) · Anonymous Referee #2 · 20 Aug 2020

This paper presents an experimental test of a method for generating estimates of characteristic wind properties based on loads measured on the turbine blades. Overall, I have no major objections to the methodology used or the results presented—it is refreshing to see data from a previous campaign being recycled and used to produce new results. However, I have some more comments and suggestions that I feel could improve reader understanding, as follows:

1. Please provide a description (in text) of shear and misalignment in Section 2.1. The equations are not directly intuitive, and Figure 1 does not actually demarcate the quantities kappa_v, kappa_h, chi, or phi. Please include units for these, where appropriate. Similarly, please provide a reference or further explanation for eqs. (3a) and (3b), since it is not immediately obvious to the reader why one involves both chi and phi and the

other does not.

2. I find Section 2.3.2 to be confusing to read. Can you be more precise about what the power law is adding? On page 9, line 16 you say that it is 'useful' to fit a power law, but don't explain why. Since using the power law seems to make the procedure of calculating vertical shear more complicated than horizontal shear, perhaps you could provide a diagram of the steps needed to calculate each state, or a concise list of steps.

3. In Section 3.2, is the SEWS being used to generate the reference value for horizontal shear, given the lack of met mast measurement? This should be clarified. If so, it would seem that the error metric for kappa_h is a difference between two competing methods as opposed to an error, which you allude to on page 16, lines 4–5, but isn't clearly stated. It seems a little inconsistent to provide results for horizontal shear but not upflow angle, since the met tower did not provide data on either.

4. Please provide better evidence for the third conclusion you draw in Section 4, regarding the improvement in the quality of training data when averaged (page 18, lines 13–14). This is an interesting result, but the only other mention of this that I found was a statement on page 15 (lines 13–14), which seem a bit brief to lead to a conclusive result.

I also have the following minor comments:

- Page 3, lines 20–25: Do you have a suggestion of how to get such an ideal measurement? I think that you're being a bit hard on yourselves, no measurement is perfect!

- Page 6, line 4: Equation 8 (the weighted least-squares solution) should have a bit more explanation. Is Q known? How did you measure/approximate it?

- Page 5, line 20: Can you provide a reference for the statement that horizontal shear varies less than vertical shear?

- Page 12, lines 1–2: Was that shift not done for the shear, also? Why shift the misalignment measurement but not the shear measurement?

---

## Author Comment (AC1) · 28 Sep 2020

**Reply to Reviewers**

We thank the reviewers for their detailed analysis and constructive inputs. A list of point-by-point replies to the reviewers' comments is detailed in the following.

**Reviewer 1**

*A good account of very careful, detailed work on a technique which could have useful applications, with a good attempt to account for the inevitable difficulties of a real field test. Detailed comments (P=page, L=line):*

1. **Reviewer**: *P2 L4: the term "lower spectrum" is not clear - or is it a typo?*
   **Authors**: We meant the lower frequencies (such as 1 per revolution, or 1P). The text was updated for improved clarity.

2. **Reviewer**: *P3: in L2 "does not need to be trained with data" but in L7 "is tuned" - what sort of tuning (and isn't that a simple form of training with data)?*
   **Authors**: The observer does not need to be trained with field data since it can be derived from the standard performance curves of the machine. Nevertheless, to compensate for possible sources of error, it can be fine-tuned with operational data. The text was updated for improved clarity.

3. **Reviewer**: *P4 L8: change "write" to "can be written as" or "are given by"*
   **Authors**: This has been changed.

4. **Reviewer**: *P4 Figure 1: top left figure (vertical shear) shows arrows originating from a non-vertical line. Does this represent rotor tilt? It should be explicitly stated somewhere that rotational symmetry depends on the coordinate system being tilted to align with the shaft axis. Presumably, after estimation you would need a final step to transform the wind components back to the 'normal' (untilted) reference frame.*
   **Authors**: As suggested, the text was rephrased to explain that the reference frame in Fig. 1 is a nacelle-attached one, considering the nacelle uptilt.

5. **Reviewer**: *P5 L21: "horizontal shear does not (except in waked conditions)" - probably not very much on average, I agree, but still maybe a bit sometimes, at least onshore due to orographic / vegetation effects - but anyway there can be significant, though short-lived, stochastic changes in horizontal (and vertical) shear across the rotor due to spatial turbulence effects.*
   **Authors**: Thank you, indeed the text was not very clear and possibly misleading, and it has now been modified to better explain this point.

6. **Reviewer**: *P6 L4: I think you should explain how Equation (8) is derived, and where Q comes from.*
   **Authors**: The text was changed to better explain the formulation; a new bibliographical reference on weighted least squares estimates was also added for completeness.

7. **Reviewer**: *P7 L5: "fairly robust to changes" - maybe "fairly robust to typical changes"? Presumably can't be true in the case of large changes (hopefully unlikely).*

**Authors**: The sentence was changed as suggested.

8. **Reviewer**: *P7 L27: Using phase-shifted measurements from blade 1 & 3 to estimate the load on blade 2: Presumably this actually means time-shifted using rotor speed? Probably needs an equation here, and a some justification for the absence of a blade 2 measurement not affecting the results. Can you say why there wasn't a measurement on blade 2?*

    **Authors**: The measurement of the load components of blade 2 was unfortunately not available, simply because that blade was not equipped with load sensors based on a choice of the turbine owner and operator. To apply the Coleman transformation, the load on blade 2 is necessary and, as explained, was obtained by averaging the loads of blades 1 and 3, which is probably the only way this load can be reconstructed with the present setup. This approach implies the reasonable assumption that neighboring blades are experiencing similar loads at the same azimuthal position. The shift is due to the angular spacing of the blades, so it is best described in terms of angles as done here; clearly, since $\psi = \Omega t$, the map between time and angle depends on angular speed.
    We have rephrased this part of the text to better explain these aspects of the discussion and improve clarity.

9. **Reviewer**: *P9 L12: presumably the reference explains in more detail, but can you give a justification for using 2/3 R? Wouldn't it depend on the turbine aerodynamic details?*

    **Authors**: We have rephrased this part, explaining that this result is valid for a linear shear and a 90-deg-wide sector. The reference where this result was first derived is also included.

10. **Reviewer**: *P10 equations 16 & 17: Consistency of notation: is VPL(z) the same as V(z)PL?*

    **Authors**: Yes, thank you for spotting this. Equation 16 was changed accordingly.

11. **Reviewer**: *P10 L10: why was a 50-degree sector chosen, and why is it not exactly centred on the direction separating mast and turbine?*

    **Authors**: The size of the sector was initially chosen to include as many points as possible while still maintaining a reasonable alignment. The sector was now changed to be exactly symmetric, which has led to a slight decrease of the number of data points.

12. **Reviewer**: *P10 L13: "two measurements" - presumably these are actually estimates rather than measurements?*

    **Authors**: The word "measurement" was replaced with "estimates".

13. **Reviewer**: *P10 L14: Since the shear is actually non-linear, and different parts of the blade contribute differently to loading, some sort of weighted mean shear gradient might be more appropriate than the slope between hub and blade tip?*

    **Authors**: The observer can be formulated to work with non-linear or linear shears. Here, the linear vertical shear was chosen in order to exploit the rotor symmetry; this has now been better explained in the text.
    A weighted shear gradient was already investigated in *"Bertelè, M., Bottasso, C.L., Cacciola, S., Daher Adegas, F. and Delport, S.:Wind inflow observation from load harmonics,Wind Energ. Sci.,2, 615–640, doi:10.5194/wes-2-615-2017, 2017",* where span-wise weighting was used to account for the non-uniform power extraction characteristics of rotors. Nonetheless, this approach did not lead to any significant difference in the profile characterization.

14. **Reviewer**: *P10 L16: different slopes in Figure 4: Might this be due to non-linearity of shear, and/or the use of 2/3 R, or do you have some other explanation?*
    **Authors**: We have provided possible explanations for this behavior, although a certain proof of the cause is difficult to achieve with the present dataset. We have modified this part of the text, which is hopefully clearer now.

15. **Reviewer**: *P11 L14: What filter characteristic, and how was it chosen? Are you filtering the wind vane signal, or the wind direction obtained by combining the wind vane signal with the nacelle position signal?*
    **Authors**: The filter is a simple 1-min moving average, applied to the difference between the wind direction measured at the mast and the nacelle position. The text has been rephrased here for improved clarity.

16. **Reviewer**: *P12 L8/9: Are you effectively assuming zero upflow, as you can't measure it?*
    **Authors**: No, since we are using the symmetry conditions for the coefficients, we do not need to assume any specific value for the upflow as it is not used for training.

17. **Reviewer**: *P13 L6: "scheduled as functions of the rotor-effective wind speed" - Not clear how you did this - was it by binning results in wind speed bins according to 10-minute average rotor-effective wind speed estimate and fitting model parameters for each wind speed bin?*
    **Authors**: The text has been improved for clarity, providing more detailed information on the implementation.

18. **Reviewer**: *Conclusions: "Training with 10-min data improves the quality of the estimates" stated without providing evidence. Can this be substantiated a bit better?*
    **Authors**: Thank you, this was indeed not documented in the previous version. We have now added a new figure (number 8 in the new version of the manuscript), and expanded the text in section 3.2 to discuss the effects of time averaging.

19. **Reviewer**: *The importance of veer is becoming more apparent especially for large turbines and stable conditions. How easily could the model be extended to provide an estimate of veer?*
    **Authors**: The current model formulation would not allow for the estimation of veer. Indeed, we have verified that, to distinguish the effects of veer, one needs to include also the 2P harmonics. These harmonics, however, are strongly polluted by turbulence and exhibit a non-linear behavior. We are currently developing a different approach to deal with these problem with promising results, which we hope to report soon in a future paper. A short comment on veer has been added to the introduction.

20. **Reviewer**: *Some comment on how the model should be adjusted in case of a turbine which is using individual pitch control?*
    **Authors**: We have investigated this issue in a simulated environment in *"Bertelè, M., Bottasso, C.L., Cacciola, S.: Simultaneous estimation of wind shears and misalignments from rotor loads: formulation for IPC-controlled wind turbines, J. Phys. Conf. Ser., 1037 032007, doi:10.1088/1742-6596/1037/3/032007, 2018"*. We added a sentence to the text to redirect interested readers to this reference.

*This paper presents an experimental test of a method for generating estimates of characteristic wind properties based on loads measured on the turbine blades. Overall, I have no major objections to the methodology used or the results presented—it is refreshing to see data from a previous campaign being recycled and used to produce new results. However, I have some more comments and suggestions that I feel could improve reader understanding, as follows:*

1. **Reviewer**: *Please provide a description (in text) of shear and misalignment in Section 2.1. The equations are not directly intuitive, and Figure 1 does not actually demarcate the quantities kappa_v, kappa_h, chi, or phi. Please include units for these, where appropriate. Similarly, please provide a reference or further explanation for eqs. (3a) and (3b), since it is not immediately obvious to the reader why one involves both chi and phi and other does not.*
   **Authors**: The picture was updated to clearly indicate the parameters and their units, and the text was expanded to better explain the relevant terms and equations.

2. **Reviewer**: *I find Section 2.3.2 to be confusing to read. Can you be more precise about what the power law is adding? On page 9, line 16 you say that it is 'useful' to fit a power law, but don't explain why. Since using the power law seems to make the procedure of calculating vertical shear more complicated than horizontal shear, perhaps you could provide a diagram of the steps needed to calculate each state, or a concise list of steps.*
   **Authors**: Thank you, indeed this part was not very clear. The text has now been partially rewritten and expanded to better explain this point.

3. **Reviewer**: *In Section 3.2, is the SEWS being used to generate the reference value for horizontal shear, given the lack of met mast measurement? This should be clarified. If so, it would seem that the error metric for kappa_h is a difference between two competing methods as opposed to an error, which you allude to on page 16, lines 4–5, but isn't clearly stated. It seems a little inconsistent to provide results for horizontal shear but not upflow angle, since the met tower did not provide data on either.*
   **Authors**: Yes, the SEWS is used to generate a reference for the horizontal shear. This is the best that can be done with the present setup and dataset. This limitation of the study was already clearly stated and explained at the end of section 3.1 "Wind shears", and in section 3.2 "Wind observer performance". The same limitation has now been added also to the introduction and conclusions. Furthermore, we have stressed the incompleteness of the dataset multiple times, warning the reader that this prevents a full validation. We believe that we honestly and openly present what we have done, and that we have amply clarified the limits of the work.
   The reason horizontal shear is discussed whereas upflow is not is simply due to the fact that for the former we have at least the SEWS, while for the latter we have nothing. Additionally, we write: "the horizontal shear is based on the same sector-equivalent wind speeds that estimate the vertical shear with good accuracy, so that there is no reason to believe that Eq. (15) should not provide a similarly good-quality estimate." Therefore, with the caveats above, a comparison of the horizontal shear with the SEWS is not completely meritless.

4. **Reviewer**: *Please provide better evidence for the third conclusion you draw in Section 4, regarding the improvement in the quality of training data when averaged (page 18, lines 13–14). This is an interesting result, but the only other mention of this that I found was a statement on page 15 (lines 13–14), which seem a bit brief to lead to a conclusive result.*

**Authors:** Thank you, this was indeed not explained in the previous version of the paper (see also a similar remark from Reviewer 1, question 18).

5. **Reviewer**: *Page 3, lines 20–25: Do you have a suggestion of how to get such an ideal measurement? I think that you're being a bit hard on yourselves, no measurement is perfect!*
   **Authors**: We have added a sentence to the introduction, indicating that a lidar scanning the inflow immediately in front of the rotor disk plane might be a possible –although still challenging– solution to this problem.

6. **Reviewer**: *Page 6, line 4: Equation 8 (the weighted least-squares solution) should have a bit more explanation. Is Q known? How did you measure/approximate it?*
   **Authors**: The text was modified to better explain this point, also in response to question 6 of Review 1.

7. **Reviewer**: *Page 5, line 20: Can you provide a reference for the statement that horizontal shear varies less than vertical shear?*
   **Authors**: This was misleading, and that sentence has been partly re-written, also in response to question 5 of Reviewer 1.

8. **Reviewer**: *Page 12, lines 1–2: Was that shift not done for the shear, also? Why shift the misalignment measurement but not the shear measurement?*
   **Authors**: Yes, all measurements derived from the mast, vertical shear included, were shifted in time. The text was modified for improved clarity.

We have taken the opportunity to make several small editorial changes to the text, in order to improve readability. A revised version of the manuscript is attached to the present reply, with the main changes highlighted in red (deletions) and blue (additions).

Best regards.
The authors

[revised manuscript text omitted]

---

## Editor Decision (ED1)

2020-83

Overall
- The subject matter of the paper is interesting but the article itself needs substantial work still
- I recommend a near full re-write of the abstract and introduction (see detailed comments below). The substantive elements are glossed over, too much vague language is used, and arguments as to the need for this approach are not sufficiently developed (see detailed notes below).
- Another major weak point is that the key results of the paper occupy a very small percentage of the overall text content of the paper – there is a lack of interpretation and explanation of the results versus describing.

Abstract
- Language in the abstract regarding the methods are somewhat vague. Overall the abstract is quite short. More specificity can be added so that the reader can have a better sense of the overall article content and impact
- Similarly for the results, the discussion of good quality / reasonable accuracy are vague terms. Falling short of real validation also vague. This can be improved quite a bit

Introduction
- Again, first sentence vague. First attempt of what? This particular sensing method? Any method using load harmonics? It is an odd way to start the paper. Usually you would start with the larger motivation and need, state of the art and then build to what this paper is going to do and its novelty at the end of the introduction.
- Again, discussion on wind sensing and rotor response with blade load sensing is a bit vague – what are the common sensor types? What blade load sensor types are you specifically referring to? How often are the actually available in standard practice at commercial farms? From what I understand, additional sensors for blade loading are not commonly applied in commercial practice. The most we typically have in a commercial park is the scada.
- "In a nutshell" is colloquial language, avoid such language in scientific writing – the explanation is weak of how the overall method works.
- I don't follow the logic of the bullets, or the argument here. What is the argument you are trying to make?
  - "Deterministic" is not the right terminology, what do you mean to say here?
  - Lower not slower sampling rates, or you can say less frequent sampling – but this is only important if 1P is a more important load to measure versus other harmonics
  - Explain the third bullet – why is this a good thing?
  - Yes, but if these harmonics and associated loads aren't the most critical design driving loads then none of this matters
- What do you mean polluted by turbulent eddies? Polluted is again colloquial verbiage and doesn't characterize scientifically what is going on – and is veer the only other characteristic we are interested in?

Overall for the first page and a half, there is a lack of a good argument as to what the different methods are that are out there, why going after the 1P makes sense and what you get/don't get by going after that measurement

- Pg. 2 line 8, Jumping to the load-harmonic method and data training without fully explaining what it is and how it compares to other methods
- Discussion about method applied in simulations or in datasets is odd. The use of it in a simulation environment would be to explore the physics and determine the feasibility of the method. The use of it in the field would be to show experimentally that the method works (i.e. validate)
- Explain why you can trust the method enough on its own without a met-mast in subsequent usage for other turbines.
- Aspects of implementation is better than implementational aspects
- Page 3 lines 1-3, the methods themselves are still not well explained and now a second is introduced without proper explanation
- Lines 10-17 – this is the first time this type of approach is used correct? How does it differ from what has been done in the past (be more explicit)
- Good discussion of limitations of the method. Make sure to circle back to it in future work
- Rather than using a "true validation" terminology, this should be seen as a field demonstration. Speak to what you do validate – what can you say from the results of the analysis that are novel and interesting? "interesting and very promising insight" is again vague – what do you get out of this study?
- Do not speak to your opinion in a scientific paper. Remove that statement.

Methods

- 2.1
    o First paragraph and Fig. 1 are very basic concepts it could be made smaller with all 4 images on one line. Put the vertical shear and uplow next to each other and then the yaw and horizontal next to each other. Why is there a slight tilt in the line for vertical shear? It looks slightly odd.
    o 10 minute averaging for wind energy applications is used often due to the characteristic frequency content in the wind itself
- 2.2
    o "in a nutshell" used again, review full paper to remove such casual language and phrases – replace that language with a more full and clear explanation.
    o Is it true that the the wind misalignment and vertical shear / horizontal shear affect loads in a symmetric fashion? There is evidence out there in a number of studies that this is not the case. It is okay to make a simplification for the sake a of study, but be caserful about what is claimed as "true." See for example: https://wes.copernicus.org/articles/3/173/2018/wes-3-173-2018.pdf
    o The whole discussion around shear and veer characteristics related to physical features and wind phenomena and the tie to rotational symmetry could be much stronger

- 2.3
  - As already mentioned, the whole argument around being able to generalize the observer design for turbines of the same type once developed for one is insufficiently explained / developed
  - How was robustness of the method shown? I assume model-based efforts were involved since this is the first field demo? And when you say method, which method are we talking about? Earlier you suggested you were using two methods together in this study
  - Here is the first mention on page 8 line 12 of the actual load sensors being used and how they are set up, there should have been some discussion on this much earlier
  - Can you speak to the limitations of the approach for averaging the loads for blade 2? When shifting the loads of blade 1/3 where there any significant deviations? – the next paragraph mentions this specifically
  - The scaling of the measurements is as specified with this factor s does not seem well-grounded since it essentially assumes that the two sensors are off by an equivalent but opposite bias. Since this is a demonstration of method, it is okay to do these sorts of things, but it needs to be explicit that this was done due to limitations of the experimental set up and is an area for future work – alternatively, the sensors could be inspected after the fact to assess their calibration status
  - The explanation for not using the wind vane is also not strong. There is indeed bias and uncertainty with win vane sensors. But saying they are off (without reference or qualification) is a weak argument. An easy excersise to correct for bias is to inspect the 0 to 360 wake profile of the turbine and see if the wake from the other turbine is where you expect it to be…
  - "in a nutshell used again, pag 8 line 31" remove hat and explain fully what you mean.
  - Bottom page 11 and top of page 12 – how much data did you have in the study overall? How long was the experimental campaign? It seems like there is something missing in terms of the overview of the campaign and how much data you have. I assume here that in the results in Fig 4, that you are using all the data you have and not accounting for different stability conditions etc that would affect the shear profile differently. You could bin the data by TI (low, moderate, high) if you have enough of it and see how well the shear profile matches under those conditions. In the right-hand side of figure 4, there seem to be significant outliers even though the overall R2 is still quite high
  - Again on the nacelle yaw sensor bias, inspection of the turbine wake location from the upstream turbine can help. Comparing two similar sensors requires assuming one is truth which is problematic unless direct calibration of one of the sensors is done before the experiment (which is always a good idea though costly)

Results
- The meat of the paper is in figures 10 through 12 with corresponding text beginning on page 17 line 6. Only 17 lines of text are dedicated to these results and the text is

descriptive (rather than interpretive). Too much attention is given to the site description and way to little attention to the actual analysis and interpretation of the results. Explain WHY the method does better under different conditions than others, what do you see as the main impact of the results? What are the key limitations? Some of the introduction discussion of limitations could be brought in here and discussed within the context of the results found.

- Tying the results back to the underlying physical phenomena, models, experimental set up and the triangulation of the 3 to explain what you understand and what the study tells you is critical to establishing the scientific value of the paper.

Conclusions
- Revisit the conclusions once the rest of the paper updates are made. A lot of the previous comments also apply here.
- Strengthen the overall closing statements

---

## Author Response (AR2)

**Reply to Editor and Reviewer**

We have received by email additional comments from the Editor and another Reviewer, which were not posted online on the journal web site. We provide here a list of point-by-point replies.

**Reviewer**

1. **Reviewer**: From a manufacturing perspective, it is easier to install sensors on the tower. In this case the procedure should be based on the 3P, rather than the 1P.

**Authors**: In theory the reviewer is right, but in practice this is not possible. There are two reasons for this:

- We have shown in several papers (all cited in the current article), that the reconstruction of the four wind states requires the 1P harmonics of both the out and the in-plane loads. While the 1P out-of-plane harmonics could in principle be reconstructed from the fixed frame OP, this is not possible for the in-plane harmonics.
- Beyond the OP, the next fixed frame harmonic is the 3P, which is generated by 3P blade loads. In Wind Energ. Sci., 2, 615–640, doi:10.5194/wes-2-615-2017, 2017 we have shown that harmonics higher than the 1P are more strongly affected by turbulence than the four wind states (shears and directions). The pollution caused by turbulence hinders the observation of the wind states.

Because of these reasons, this wind estimation method should be formulated in terms of rotating measurements.

However, in our opinion this is not a limitation of the method. While it is true that installing sensors in the fixed frame is easier from a manufacturing perspective, it is also a fact that blade sensors are becoming more and more common: many of the more recent turbine platforms come equipped with blade sensors, and many retrofitting options for existing turbines are available on the market today. In fact, as the price of these sensors falls, it is becoming more and more appealing to install them in support of load-reducing control and –even more commonly nowadays– for the continuous live monitoring of loads in support of maintenance, fault detection, lifetime consumption estimation etc. As stated in the paper, once these sensors are already installed on a turbine, wind sensing is a simple software upgrade, i.e. it is a new function that reuses data already available to provide extra information on the operating ambient conditions at no cost.

We have taken the opportunity of this comment for expanding the discussion on this point in the manuscript.

2. **Reviewer**: The turbulence will still affect the rotor speed, and in turn the 1P. If the measuring time is too long, the peak of the 1P will get broader.

**Authors**: Measuring time is not too log: as stated in the paper, all measurements are acquired at 10 Hz, and the harmonics are produced by the Coleman transformation at the same rate. This means that one has a new "wind state—load harmonic" pair each tenth of a second. Since we use 1P harmonic amplitudes, we automatically accommodate for variable rotor speeds. The only assumption is that the rotor speed does not change too much from one sample to the next, i.e. within one tenth of a second, which is a pretty good approximation.

The idea that "the peak of 1P will get broader" refers to a typical spectral diagram, which is not what we are doing here.

- 3. **Reviewer**: *This is a huge advantage over OMA.* **Authors**: Indeed, it is.
- 4. **Reviewer**: *Of which rated power? This is very type-specific and should be removed.* **Authors**: We agree that the comment is too specific, and it has been eliminated.
- 5. **Reviewer**: But especially because the energy contained in the wind decreases with the frequency. **Authors**: That is absolutely correct. This is also a good argument for considering the lowest frequencies of the machine response when trying to reconstruct the wind inflow. The sentence has been modified accordingly.
- Reviewer: This links to "...or be generated synthetically using a simulation model".
   Authors: Exactly. If we can rely on the fact that simulation tools can capture reasonably well the lowest frequency response, we can also rely on them to identify the observation model starting from simulation data.
- 7. Reviewer: Also the veer is deterministic, and as such should be visible without turbulence. Authors: The veer is indeed deterministic, and in principle it should be observable. In fact, we have tried this already, but so far without much success. Our numerical simulations show that veer influences not only the 1P, but also the 2P. Unfortunately, 2P harmonics are also strongly affected by turbulence (*Wind Energ. Sci., 2, 615–640, doi:10.5194/wes-2-615-2017, 2017*). Therefore, the estimation of veer in turbulent wind conditions becomes very uncertain and imprecise. We are still investigating this aspect, but at the moment we cannot claim to be able to observe veer.

**8. Reviewer: Mean and amplitude?**

**Authors**: As described in detail in "*Bertelè*, *M.*, *Bottasso*, *C.L.*, *Cacciola*, *S.: Simultaneous estimation* of wind shears and misalignments from rotor loads: formulation for IPC-controlled wind turbines, *J. Phys. Conf. Ser.*, 1037 032007, doi:10.1088/1742-6596/1037/3/032007, 2018", the 1P pitch inputs are considered both in phase and amplitude. The sentence has been modified.

- Reviewer: Considering that for zero shear we get W = V\_h, the notation is quite confusing. I guess that W and V\_h are wind velocity magnitudes.
   Authors: We do not believe the notation to be confusing. A "velocity magnitude" in English is called "speed". W is the speed bi-linear function defined in the y-z plane, and for y=z=0 its value is equal to V\_h. Additionally, if the shears are zero, the wind speed is constant at each z and y position over the rotor disk, and equal to the one at hub height. Exactly as the reviewer suggested.
- 10. **Reviewer**: *Rewrite as: "The rotor loads are assumed to be linearly dependent on the wind states"* **Authors**: The sentence has been changed.
- 11. **Reviewer**: The load vector *m* is constant only if both the rotor and the external conditions are isotropic. Otherwise, if the wind is steady, it is azimuth-dependent. If the wind is turbulent it will be time-varying.

**Authors**: What stated by the reviewer is true for the load when seen as a function of azimuth and/or time, but it is clearly not correct for the load harmonics. This is actually the exact reason why we use load harmonics.

- Reviewer: This load vector does not include the collective in-plane and out-of-plane components. I'm unsure if this is embedded in m0, but it should be clarified.
   Authors: If by "collective" the reviewer means the OP component of the flap and edge-wise moment, then of course these terms are not included. Clearly, the collective OP does not include any information on the shears nor the directions. We do not believe this needs further clarification in the text, since we already clearly state that the vector contains "1P sine and cosine harmonic amplitudes".
- 13. **Reviewer**: *If m contains only the 1P component, then the low pass filter must filter all the other nP, as well as the turbine modes. Further details are needed.*

**Authors**: As described in the paper, the 1P is extracted via the Coleman Transformation, and then a low pass filter is applied. As explained in the reference provided, the n-th order Coleman Transformation will shift the frequency of the desired nP to a corresponding 0P, whereas any other frequency is either canceled out or shifted to higher harmonic terms. Therefore, a simple low pass filter suffices to extract the 0P harmonic of this transformed signal, which corresponds to the 1P harmonic of the original blade loads. We do not believe additional details are needed, since the interested reader is provided with a reference and this is standard text book material.

14. **Reviewer**: Here the authors are not considering the bending of the tower, caused by a vertical shear. This will affect the estimate of the upflow angle.

**Authors**: This and other deformation-induced effects are taken into account by scheduling the wind-load model by density and wind speed. Section 2.2 has been partially modified to better explain this fact, including the example given by the reviewer.

15. **Reviewer**: I don't like this expression, because it is not the actual implementation (I hope). It suffices to say that Eq. 6 is solved in the least squares sense.

**Authors**: One should not confuse the mathematical solution of a problem with its numerical implementation. The expression reported in Eq. 7 is the correct mathematical solution of the problem described in Eq. 6, which is a very classical text book result. Probably the reviewer does not like the inversion of matrix  $\Theta\Theta^T$  that, if performed naively, can cause numerical precision issues. This is however common knowledge, and we do not think it requires a dedicated explanation, otherwise one could never write a matrix inverse in a paper.

16. Reviewer: The residual must also be white, and hence its covariance matrix must be an identity matrix. Otherwise, the solution of the least squares problem (6) will be biased. Inside the unmodeled physics there is also the atmospheric turbulence, which is definitely colored. Authors: We perfectly agree with the reviewer, and in fact we do not claim that the estimates are unbiased. Therefore, the covariance is not an identity matrix. This is however standard material and a classical result in the presence of colored noise. We have added a reference ("R.V. Jategaonkar: *Flight Vehicle System Identification: A Time-Domain Methodology*, Second Edition, ISBN:978-1-62410-278-3"), which covers well this topic.

17. **Reviewer**: "online" normally means that the estimate is updated by adding new measures to the old ones, but this does not seem to be the case here.

**Authors**: We disagree that online has the meaning of updating old measurements with new ones. A search in the main dictionaries shows that the most common acception of this term is the one used by us, i.e. "done continuously as the system is operating", in opposition to offline.

18. **Reviewer**: If the purpose of this expression was to introduce an extremely non-linear, nondimensional, coefficient C, then the authors should check the dimensions. In fact, this expression is wrong for bending moments (Nm on the left and N on the right hand side). I would also like to stress that C is very dependent on several parameters, like the pitch angle and load channel. Even the density, which appears linearly in this expression, is still contained in C. The same goes for the wind speed.

**Authors**: We thank the reviewer for noticing this issue with dimensions, which has now been corrected. We do agree that *C* depends on several parameters. In fact, in *Wind Energ. Sci., 2, 615–640, doi:10.5194/wes-2-615-2017, 2017* we have defined the term as scheduled with respect to density and wind speed (which, in turn, implies also a dependency on pitch angle). In the present description, we had not explicitly remarked this dependency to simplify the discussion. However, to avoid misunderstandings, we have now modified the text.

- Reviewer: If the rotor is isotropic.
   Authors: Of course, we agree with the reviewer. We have modified the text accordingly.
- 20. Reviewer: The 1P is the major effect of the gravity, but not the only one, since other nP will be present. Is this also addressed with the filtering?
  Authors: We agree that, although gravity might have smaller effects at higher nPs, it will mainly influence the 1P frequency. However, we do not understand why this should be an issue for the filtering. Since we are interested only in 1P harmonics, as very clearly stated in the paper, we do filter out all remaining contributions.
- 21. **Reviewer**: *This is again dimensionally wrong for bending moments.* **Authors**: Thank you for pointing this out. We have corrected the text.
- 22. **Reviewer**: The gravity loading depends on the pitch angle and blade deflection. Thus, it cannot be identified only at the cut in

**Authors**: We disagree with this comment of the reviewer. Section 2.2.1 explains that gravityinduced harmonics can be split in two contributions: a first term that represents gravity-induced loads due to the rotor deformation caused by aerodynamic loads; and a second term that represents gravity-induced loads that depend on the initial undeformed configuration, and hence do not depend on density. The goal here is to estimate this second term; hence, it makes sense to do this at cut-in.

23. **Reviewer**: If the aerodynamic loads are negligible at the cut in, then there is little point in starting the turbine. The authors should quantify this statement.

**Authors**: It is not that aerodynamic loads are negligible, but, as stated in the text, the term that is negligible is the "gravity-induced load due to the rotor deformation caused by aerodynamic loads". This indeed is small, as we have verified with the help of aeroelastic simulations.

In addition, we do not understand the statement "... *then there is little point in starting the turbine*": the turbine will start anyway, since this is what it does when the wind speed reaches the cut-in value. Here we are simply using data measured in these conditions to estimate the gravity term of the equation.

- 24. **Reviewer**: *Why 60 deg instead of 120? Is it a typo?* **Authors**: It is a typo, thank you for pointing this out.
- 25. **Reviewer**: The strain gauges will undergo bias and scaling issues. It's impossible to correct for this problem without a reference. The functions m1 and m3 are unclear. How is determined s? Writing this number does not provide any useful knowledge to the reader. Is the calibration independent on the load channel?

**Authors**: It is indeed very difficult to correct for sensor bias without proper references or direct access to the machine and to the measurement equipment. The data used in this validation was recorded in 2017. Given that the authors have acquired the dataset years later, unfortunately the data can only be analyzed a posteriori without references or additional information. Therefore, the blade loads were just rescaled to ensure that the long term mean of the measured loads on blade 1 and 3 are consistent, and that is exactly how parameter *s* is computed. Of course, this is just an a posteriori correction that by no means implies that the sensor readings have been properly recalibrated with respect to the (unknown) real values. This correction was applied on the root-bending moment channels, which were also the only load channels available. The text has been slightly rephrased to make this point clearer. Additionally, we have now added both in the introduction and in the body of the paper that the dataset was collected prior to the present study, which should make it clear to the reader that we had only limited options for data improvement and cleaning.

26. **Reviewer**: Cone is normally associated to the average flapwise moment, while here each blade will produce a different one

**Authors**: As described in the definition, the cone coefficient depends on the azimuth angle. So it is correct that, at each instant of time, each blade will have its own cone coefficient. This is indeed what allows one to use each blade as individual sensor, as done by the SEWS observer of *"Schreiber, J., Bertelè, M., and Bottasso, C.L.: Field testing of a local wind inflow estimator and wake detector, Wind Energ. Sci., 5, 867–884, doi:10.5194/wes-5-867-2020, 2020"*. In the same reference it is also shown how the cone of the rotor can be derived by averaging, at each instant of time, the values of the individual blade cone coefficients.

27. **Reviewer**: This makes the approach model-based, and contradicts a previous statement.

**Authors**: The approach described in this sentence is not the one under validation, but another already validated one (SEWS, "Schreiber, J., Bertelè, M., and Bottasso, C.L.: Field testing of a local wind inflow estimator and wake detector, Wind Energ. Sci., 5, 867–884, doi:10.5194/wes-5-867-2020, 2020") used as reference. We believe that the use of two observers is described very clearly in the paper, and should not be missed or misunderstood by a careful reader. In fact, we spend a good part of the introduction just to explain this point, which is then explained again in 2.3.2 and in the conclusions. The text was slightly modified to clarify this fundamental difference even more.

28. **Reviewer**: It's a bit strange to dedicate half of the paper to rotor quantities, and suddenly start looking at each blade.

**Authors**: As for the previous comment, this is not the description of the methodology under validation. This is the description of an already validated tool (*"Schreiber, J., Bertelè, M., and Bottasso, C.L.: Field testing of a local wind inflow estimator and wake detector, Wind Energ. Sci., 5, 867–884, doi:10.5194/wes-5-867-2020, 2020"*) that we can use to derive rotor effective references for the load-harmonic observer. We believe this is very clearly and extensively explained in the paper.

29. **Reviewer**: So, we first have to estimate a nonlinear shear and then linearize it? Why loosing accuracy? Why not determine the linear shear, then linearize Eq. (17) at hub height, and finally solve for alpha?

**Authors**: The harmonic-based method estimates a vertical linear shear (see figure 1). This has the advantage of using the same definition for the horizontal and vertical shears, which allows for a simplified formulation (with fewer unknown parameters) based on the symmetry of the problem (*"Bertelè, M., Bottasso, C.L. and Cacciola, S.: Wind inflow observation from load harmonics: wind tunnel validation of the rotationally symmetric formulation, Wind Energ. Sci., doi:10.5194/wes-2018-61, 2019"*).

Therefore, since the observed shear is linear, it has to be compared with other similarly defined shears obtained by the met mast and the sector-effective observer. As explained in the paper, this is not obvious, because all these various shears are obtained from a different number of measurements at different heights above the terrain. The procedure that we have formulated, and explained in detail in the paper, is the one that in our opinion is the most accurate for performing this comparison. The text has been modified to better explain this point.

30. **Reviewer**: It was written earlier that the wake operations where excluded. A full wake might be acceptable for this model, but not a partial wake. The authors should clarify this.

**Authors**: In section 2.3.1 we have stated that waked conditions were discarded from the training data set. But this conditions can and were included in the validation data set.

We do not agree with the reviewer's comment "a full wake might be acceptable for this model, but not a partial wake". A partial wake results, in addition to other effects, into a horizontally sheared flow. Indeed, both the harmonic and the sector-effective observes do estimate horizontally sheared inflows. Therefore, such conditions are perfectly acceptable for the proposed model. Indeed, the main reason for the inclusion of the horizontal shear in the observed parameters is to detect wake impingements (see "Schreiber, J., Bertelè, M., and Bottasso, C.L.: Field testing of a local wind inflow estimator and wake detector, Wind Energ. Sci., 5, 867–884, doi:10.5194/wes-5-867-2020, 2020").

- 31. Reviewer: With this method, a mass imbalance or pitch misalignment will be detected as a different inflow. Addressing this problem should be included in the future works.
  Authors: Imbalances will affect each blade differently, while the overall rotor-equivalent wind states are obtained by the combined effects of the three blades. Because of this reason, imbalances have only a modest effect on the estimates, as studied in C.L. Bottasso, C.E.D. Riboldi, 'Validation of a Wind Misalignment Observer using Field Test Data', Renewable Energy, 74:298{306, doi:10.1016/j.renene.2014.07.048, 2015. Additionally, the effects of imbalances can be mitigated by the heuristic long-term averaging used in 2.3.1, which ensure similar measurements coming from the three blades. The text has been modified to address this point.
- 32. **Reviewer**: I guess that the turbulence intensity has been included in the large "non-modeled" physics.

**Authors**: The model is trained on 10-minute averages, therefore it can be considered a "steady model" that is oblivious of turbulent effects. As shown in *Wind Energ. Sci., 2, 615–640, doi:10.5194/wes-2-615-2017, 2017, 1P* harmonics are predominantly affected by the four wind states, while turbulence affects the higher harmonics. This also explains why the observer performance does not dramatically depend on turbulence intensity.

**Editor**

**Overall**

1. Editor: The subject matter of the paper is interesting but the article itself needs substantial work still. I recommend a near full re-write of the abstract and introduction (see detailed comments below). The substantive elements are glossed over, too much vague language is used, and arguments as to the need for this approach are not sufficiently developed (see detailed notes below).

**Authors:** We are glad to use this opportunity to improve the paper, as detailed below and earlier on when replying to the reviewer. In fact, we have made many changes to the text and accommodated a very large part of the requests from the Editor and the Reviewer, as shown in the attached "diff" version of the manuscript.

However, unfortunately we have the impression that the words and tone of this review read very much like a lecture in style and on how to write a scientific paper. We respectfully disagree with the use of such a tone in a peer review. We believe that authors are entitled to their intellectual independence, and to some freedom in making their own stylistic choices.

Additionally, the timing of the many requests for deep rewritings of substantial parts of the paper is unhelpful. The same requests could have been made during the first round of reviews, since they address parts of the work that were present from the very first draft.

2. Editor: Another major weak point is that the key results of the paper occupy a very small percentage of the overall text content of the paper – there is a lack of interpretation and explanation of the results versus describing.

**Authors:** We respectfully disagree. We have a rather long initial description because we tried to explain the methodology as clearly as possible. Indeed, we had to introduce the method, the experimental setup and its limitations, the additional estimator used to provide a reference for shear; we tried to convey all this complicated information as concisely as possible, and we believe that eliminating part of the description will not improve readability. From section 2.3 onwards, we not only provide a detailed explanation of the test site and of the data, but we also analyze the data itself and the results, including: a detailed analysis of the shear including the difference between full and half-rotor shears; the interpretation and correction of wind misalignment; the analysis of wind speed correlation; the observer performance as a function of both identification and estimation sampling frequency; the performance in the time domain; the performance as a function of wind speed, turbulence intensity and wind direction. Many of these analyses have been quantified in a statistical sense in terms of correlation and error, not for one but for all three available inflow parameters.

We believe that extending the result section will only lead to a larger number of pages, but the content will remain the same, as the content itself is limited by the content of the dataset that we have used. We have stated very clearly that the data set has limitations: in the introduction, the body of the paper and the conclusions. An accurate description and interpretation of the results can be provided also with a few paragraphs, especially if the linked visual aids (i.e. figures) are

almost self-explanatory. We believe that we have made all comments that are clear and to the point.

**Abstract**

3. Editor: Language in the abstract regarding the methods are somewhat vague. Overall the abstract is quite short. More specificity can be added so that the reader can have a better sense of the overall article content and impact

Authors: We have revised the abstract.

4. Editor: Similarly for the results, the discussion of good quality / reasonable accuracy are vague terms. Falling short of real validation also vague. This can be improved quite a bit.
Authors: We have revised the text, avoiding where possible vague statements. On the other hand, we believe that in some parts of the text, for example in the introduction, it is unnecessary to be excessively precise, as the reader does not have yet enough information to appreciate (or even fully understand) specific and precise figures. It is indeed one of the goals of the introduction to explain the problem and the results in general terms, which are clearly made more precise throughout the rest of the paper. In addition, this is a style issue, which could be left to the authors' taste and preferences. In fact, precise results are indeed provided throughout our paper by table and diagrams, as in all scientific publications.

**Introduction**

5. **Editor:** Again, first sentence vague. First attempt of what? This particular sensing method? Any method using load harmonics? It is an odd way to start the paper. Usually you would start with the larger motivation and need, state of the art and then build to what this paper is going to do and its novelty at the end of the introduction.

**Authors:** Thank you for the explanation on how to start a paper. However, we respectfully disagree, as this comment relates to the writing style, which is a personal choice of the authors. We prefer to start the paper directly telling the reader what the paper is about: a first attempt at *"the field validation"* of *"a wind sensing method based on load harmonics"*. The details on what wind sensing is, the methodology etc. are mentioned in the following lines and throughout the whole paper. Therefore, the first sentence is, in our opinion, anything but vague. Actually, it is extremely precise and lets the reader know immediately from the very beginning what the paper will be discussing. In our opinion this is better than very generic introductions that start from far away, where one has to go through several paragraphs before understanding the main contribution of the article.

6. Editor: Again, discussion on wind sensing and rotor response with blade load sensing is a bit vague – what are the common sensor types? What blade load sensor types are you specifically referring to? How often are the actually available in standard practice at commercial farms? From what I understand, additional sensors for blade loading are not commonly applied in commercial practice. The most we typically have in a commercial park is the scada.

**Authors:** We have expanded the text, also based on a comment by the reviewer (see reply 1 to the reviewer's comments).

7. Editor: "In a nutshell" is colloquial language, avoid such language in scientific writing – the explanation is weak of how the overall method works.

**Authors:** The comment refers to the writing style of the paper, which is purely a subjective matter. We do not believe that expressions such as "in a nutshell" have no place in a scientific publication. Nonetheless, we have eliminated it to accommodate this request.

We respectfully disagree that the explanation is weak (which is again a subjective remark): this expression introduces an explanation that is concise and straight to the point (i.e. "some characteristics of the inflow (horizontal and vertical shear, lateral and vertical misalignment angles) generate a specific response of the rotor at the 1P (once per revolution) frequency"). The interested reader can find further information in the references immediately listed below as well as, of course, in the Methods section.

8. **Editor:** *I* don't follow the logic of the bullets, or the argument here. What is the argument you are trying to make?

**Authors:** We are trying to make the point that it is "a very desirable feature" that "some characteristics of the inflow (horizontal and vertical shear, lateral and vertical misalignment angles) generate a specific response of the rotor at the 1P (once per revolution) frequency".

- a. Editor: "Deterministic" is not the right terminology, what do you mean to say here?
   Authors: On the contrary, we believe that "deterministic" is exactly the correct word to distinguish the effects of the wind states from non-deterministic turbulent fluctuations. We have revised the text to make this point clearer, but we have not eliminated the term "deterministic", which we have also used often in other publications.
- Editor: Lower not slower sampling rates, or you can say less frequent sampling but this is only important if 1P is a more important load to measure versus other harmonics
   Authors: Thank you for noticing this typo. This comment is of course relevant, since the fact that the 1P is the most important harmonic for the method is exactly the point we are trying to make.
- c. Editor: *Explain the third bullet why is this a good thing?* Authors: The sentence has been rewritten.
- d. **Editor:** Yes, but if these harmonics and associated loads aren't the most critical design driving loads then none of this matters

**Authors:** We do not understand this comment, as this discussion has really nothing to do with critical design driving loads. Here we are talking about estimating the inflow from loads measured during operation. The point here is that: 1) the method only relies on the lowest possible harmonic, i.e. the 1P; 2) simulations tools are typically more accurate in the lower band of the spectrum, and progressively less accurate when considering higher frequencies. Although the remark on driving loads is off mark, this whole part of the text has been slightly rephrased for improved clarity.

9. Editor: What do you mean polluted by turbulent eddies? Polluted is again colloquial verbiage and doesn't characterize scientifically what is going on – and is veer the only other characteristic we are interested in?

**Authors:** On the contrary, we respectfully disagree and we believe that "polluted" gives a good and synthetic idea of what we are referring to. In addition, pollution is commonly used in scientific writing to indicate noise that obscure useful information or a solution. The 1P harmonics are strongly dominated by deterministic effects (as mentioned in the first bullet point, i.e. the wind states that we want to measure), whereas higher harmonics are also affected by non-deterministic smaller-scale turbulent fluctuations. Regarding veer, please see the reply to comment 7 of the reviewer.

10. **Editor**: Overall for the first page and a half, there is a lack of a good argument as to what the different methods are that are out there, why going after the 1P makes sense and what you get/don't get by going after that measurement

**Authors:** In our first version we had kept this discussion quite short because these topics had been explained several times in previous publications. However, the introduction has now been expanded to address this comment.

As far as to why the 1P is relevant, we believe this is very well summarized in the bullet points, especially the first one. The other necessary details are given later in the Methodology section, where we have tried to distill the essence of the formulation from the more extensive explanations given in the various previously published papers.

- Editor: Pg. 2 line 8, Jumping to the load-harmonic method and data training without fully explaining what it is and how it compares to other methods
   Authors: We have reworded this part. Still, we believe that a detailed description of the methodology does not belong in the "Introduction" section, but rather in the "Methodology".
- 12. Editor: Discussion about method applied in simulations or in datasets is odd. The use of it in a simulation environment would be to explore the physics and determine the feasibility of the method. The use of it in the field would be to show experimentally that the method works (i.e. validate)

**Authors:** We assume the Editor is referring to the discussion of page 2. The discussion about a model-driven or data-driven method does not appear odd to us. Of course a simulation environment can be used to explore the physics of the method (as was done in several of the references listed in the paper). However, this is by no means its only use. If one has a model that is good enough to capture the effects of the four wind states on the 1P load harmonics, the observer can be obtained offline by simulations and then used online in the field. The text has now been rephrased to make this clearer, and this point is also explained at length in some of the earlier publications.

13. Editor: Explain why you can trust the method enough on its own without a met-mast in subsequent usage for other turbines.

Authors: This was written in several places (page numbers refer to the previous version of the manuscript):

- Page 2: "There should be limited variability in such low frequencies among different installations of a same wind turbine type;"
- Page 7: "A similar procedure could be used to identify the observer for a specific wind turbine type. Having obtained the model coefficients, one should be able to use the same observer for other installations of that same wind turbine type. Although there is yet no direct demonstration of this assertion, it seems reasonable to assume that wind turbines of the same model will have a similar 1P response to shears and misalignment angles. Additionally, Bottasso and Riboldi (2015) showed that the method is fairly robust to the typical changes occurring in some of the wind turbine parameters across different installations of a same wind turbine type, including changes in the stiffness of foundations, orographic effects, imbalance due to pitch misalignment, miscalibration of the load sensors and changes in airfoil lift and drag due to soiling/erosion."

• Page 20: "A remaining open point is the demonstration that the method can indeed be trained on a turbine and, then, applied to another machine of that same model at another site; although this seems to be a very reasonable assumption, the evidence that this is indeed possible is lacking."

The conclusions very clearly state that this assumption still remains to be verified. The text of introduction and conclusions has now been modified, for improved clarity.

- 14. Editor: Aspects of implementation is better than implementational aspects Authors: This has been changed.
- 15. Editor: Page 3 lines 1-3, the methods themselves are still not well explained and now a second is introduced without proper explanation

**Authors:** As we have already mentioned in the previous comments, we do not believe the introduction to be the place for detailed explanations of the methods. Here we are explaining why we use more than one observer, and the reader does not need to have a detailed understanding of the methods to follow this argument. Of course all details are provided, but only later on in the "Methods" section. We believe that the two methods are clearly defined. It is also perfectly clear which one is being validated, and what is the role of the second one.

- 16. Editor: Lines 10-17 this is the first time this type of approach is used correct? How does it differ from what has been done in the past (be more explicit)
  Authors: We assume the Editor is talking about the lines 10-17 of page 3. The sector-effective estimator has been described elsewhere, as per the citations, and as clearly explained in the text.
- 17. Editor: Good discussion of limitations of the method. Make sure to circle back to it in future work Authors: OK, thanks for the advice, we will.
- 18. Editor: Rather than using a "true validation" terminology, this should be seen as a field demonstration. Speak to what you do validate what can you say from the results of the analysis that are novel and interesting? "interesting and very promising insight" is again vague what do you get out of this study?

**Authors:** As mentioned above, we do not think that specific details are needed in the introduction. Moreover, the whole introduction clearly describes what will be validated in the paper and how. There is no need to give a detailed preview of the results in the introduction: the results are described in detail in their dedicated section and, based on them, conclusions are drawn in the final one.

19. Editor: Do not speak to your opinion in a scientific paper. Remove that statement.

**Authors:** Which statement? This is a scientific paper with very precise statements, detailed analyses and quantitative information. Of course, it is perfectly acceptable to also include more nuanced generic statements and personal opinions where appropriate.

**Methods**

20. Editor: First paragraph and Fig. 1 are very basic concepts it could be made smaller with all 4 images on one line. Put the vertical shear and uplow next to each other and then the yaw and horizontal next to each other. Why is there a slight tilt in the line for vertical shear? It looks slightly odd. Authors: The reason why the vertical shear is "slightly tilted" is precisely explained and should not be missed by the careful reader: "where x is parallel to the axis of rotation (and it is therefore inclined with respect to the ground because of uptilt) ... It should be noticed that the vertical shear is customarily defined with respect to the horizontal, instead of the uptilt, direction; additionally, its profile is typically either logarithmic or expressed as a power law, instead of linear. As explained later, these choices are made here to exploit the rotational symmetry of the rotor (Bertelè et al., 2019)".

We also do not see the need to modify the order of the wind parameters in the figure, especially since this would make the figure not consistent with our previous papers. In addition, these graphical adjustments are typically done during the production process, at the light of the journal formatting and page composition, which differs from the one used during peer review.

- 21. Editor: 10 minute averaging for wind energy applications is used often due to the characteristic frequency content in the wind itselfAuthors: That is correct. This is why we are following this approach.
- 22. Editor: "in a nutshell" used again, review full paper to remove such casual language and phrases replace that language with a more full and clear explanation.
  Authors: We have now removed this expression but, as mentioned in comment 8, we respectfully disagree with the reviewer on this purely stylistic comment. Language adjustment as the one

disagree with the reviewer on this purely stylistic comment. Language adjustment as the one suggested here are typically done by the language editor during production. By the way, we have used this expression in other publications, and it has never been modified by the language editors.

23. Editor: Is it true that the the wind misalignment and vertical shear / horizontal shear affect loads in a symmetric fashion? There is evidence out there in a number of studies that this is not the case. It is okay to make a simplification for the sake a of study, but be caserful about what is claimed as "true." See for example: https://wes.copernicus.org/articles/3/173/2018/wes-3-173-2018.pdf Authors: We believe the reviewer might have misunderstood the concept of "rotor symmetry" used here. The paper that she refers to talks about a completely different topic, namely about the difference between positive and negative yaw misalignments, which is very well known (of course also to us) and it has been described in several publications.

Here we talk about a completely different and unrelated aspect: neglecting the presence of the tower, a horizontal linear shear produces the same response of a vertical linear shear, with a 90° phase shift. Similarly, a horizontal yaw misalignment causes the same response, delayed by 90°, of a vertical misalignment. This is evident by simple geometry, and has been thoroughly discussed in a peer-reviewed paper: "Bertelè, M., Bottasso, C.L. and Cacciola, S.: Wind inflow observation from load harmonics: wind tunnel validation of the rotationally symmetric formulation, Wind Energ. Sci., doi:10.5194/wes-2018-61, 2019".

The text clearly explains what we are talking about, and all necessary details are given in the cited publication.

24. **Editor:** The whole discussion around shear and veer characteristics related to physical features and wind phenomena and the tie to rotational symmetry could be much stronger

**Authors:** We believe the Editor means "discussion around shear and direction". Please see reply to the previous question. This comment is based on a misunderstanding from the reviewer.

- 25. Editor: As already mentioned, the whole argument around being able to generalize the observer design for turbines of the same type once developed for one is insufficiently explained / developed **Authors:** Please refer to the answer to Editor's comment 13.
- 26. Editor: How was robustness of the method shown? I assume model-based efforts were involved since this is the first field demo? And when you say method, which method are we talking about? Earlier you suggested you were using two methods together in this study
  Authors: The robustness of the load-harmonic method was characterized by simulations in previous cited references as a function of wind speed and turbulence. The publication Bottasso, C.L. and Riboldi, C.E.D.: Validation of a wind misalignment observer using field test data, Renew. Energ., 74, 298–306, doi:10.1016/j.renene.2014.07.048, 2015 analyzed the effects of changes in the stiffness of foundations, orographic effects, imbalance due to pitch misalignment, miscalibration of the load sensors and changes in airfoil lift and drag due to soiling/erosion.

As stated multiple times throughout this manuscript and also in the conclusions, the robustness of the method still remains an open point to be properly investigated. The final part of the conclusions section has now been reworded to make this even clearer.

Regarding the two methods, again we had very clearly explained (starting from the abstract) why we use two methods, and their respective roles. We have now revisited again the whole text, and we believe that a careful reader will be able to easily understand how the two methods are used.

- 27. Editor: Here is the first mention on page 8 line 12 of the actual load sensors being used and how they are set up, there should have been some discussion on this much earlier **Authors:** A mention of the possible sensor types has been included in the introduction.
- 28. Editor: Can you speak to the limitations of the approach for averaging the loads for blade 2? When shifting the loads of blade 1/3 where there any significant deviations? the next paragraph mentions this specifically

**Authors:** The text already included a discussion on this point: "unfortunately, however, the same load sensors were not installed on blade 2. To reconstruct the missing load components, the measurements of blades 1 and 3 were shifted by  $\pm 2\pi/3$ , averaged together and then attributed to blade 2. This approximation assumes that neighboring blades experience the same loads when they are at the same azimuthal position, which is reasonable because loads and wind states are time-averaged quantities linked by a steady load-wind model (cf. Eq. (4))."

We do not see what else could be said regarding this point. We have included in the introduction and in this section a more precise statement, explaining that the dataset was collected prior to this study, which implies that we had very few options for improving or correcting the measurements.

Regarding deviations, please see the reply to the next comment.

29. Editor: The scaling of the measurements is as specified with this factor s does not seem wellgrounded since it essentially assumes that the two sensors are off by an equivalent but opposite bias. Since this is a demonstration of method, it is okay to do these sorts of things, but it needs to be explicit that this was done due to limitations of the experimental set up and is an area for future work – alternatively, the sensors could be inspected after the fact to assess their calibration status **Authors:** As stated in the paper, these experiments were conducted three years ago, and there is no extra information in addition to what we have used. Given the situation, we do not see how things could have been done differently nor, clearly, how we could have done any inspection to assess the calibration status, as suggested. We have expanded the text to make this even more clear, although the situation was expressed clearly already in the previous version of the manuscript.

30. Editor: The explanation for not using the wind vane is also not strong. There is indeed bias and uncertainty with win vane sensors. But saying they are off (without reference or qualification) is a weak argument. An easy excersise to correct for bias is to inspect the 0 to 360 wake profile of the turbine and see if the wake from the other turbine is where you expect it to be... Authors: If the reviewer is referring to page 8, lines 23-25: we are not saying that the wind vanes

Authors: If the reviewer is referring to page 8, lines 23-25: we are not saying that the wind vanes are "off", but that they need to be carefully calibrated, which is a well-known fact. Although nacelle-mounted wind vanes are not always very precise, we have verified that in this specific dataset this sensor correlates well with the mast. However, since the two did not exhibit any significant difference, we decided to use the mast, for coherence with the other reference quantities that are also measured at the mast. The text has been revised accordingly.

- 31. **Editor:** *"in a nutshell used again, pag 8 line 31" remove hat and explain fully what you mean.* **Authors:** Done, but please see also our replies to similar previous comments.
- 32. Editor: Bottom page 11 and top of page 12 how much data did you have in the study overall? How long was the experimental campaign? It seems like there is something missing in terms of the overview of the campaign and how much data you have. I assume here that in the results in Fig 4, that you are using all the data you have and not accounting for different stability conditions etc that would affect the shear profile differently. You could bin the data by TI (low, moderate, high) if you have enough of it and see how well the shear profile matches under those conditions. In the right-hand side of figure 4, there seem to be significant outliers even though the overall R2 is still quite high

**Authors:** We are surprised by these statements, which do not seem to reflect the content of the manuscript. Referring to the previous version of the paper, page 7, lines 28-29 state: "Data was measured between October 19 and November 29, 2017 on a 3.5 MW eno114 turbine designed and produced by eno energy systems GmbH". We give a quite precise indication also of how many hours of useful data are available at page 14, lines 13-15. Also the lower left plots of Fig. 10-12 show how many hours of data are available as a function of wind speed, density, wind direction and rotor effective turbulence intensity.

As reported in the text, page 11 lines 5-7, we are not using all wind conditions, but only the data for reasonable turbine-mast alignment, and in the subsequent lines we describe possible reasons for the shown trends and outliers.

We have no information about the atmospheric stability conditions, except the one that could be derived by looking at shear and TI.

Regarding the comment on binning with respect to TI, this was in fact done and the results are shown in Figure 10.

33. Editor: Again on the nacelle yaw sensor bias, inspection of the turbine wake location from the upstream turbine can help. Comparing two similar sensors requires assuming one is truth which is

problematic unless direct calibration of one of the sensors is done before the experiment (which is always a good idea though costly)

**Authors:** We agree with the reviewer comment. It would be great to recalibrate the sensors prior to the experiments. However, since we have "repurposed" a dataset that had been collected years earlier, unfortunately we can only do the best we can with the available information and measurements. As far as the wake-location suggestion is concerned, please refer the answer to comment 32.

Results

34. Editor: The meat of the paper is in figures 10 through 12 with corresponding text beginning on page 17 line 6. Only 17 lines of text are dedicated to these results and the text is descriptive (rather than interpretive). Too much attention is given to the site description and way to little attention to the actual analysis and interpretation of the results. Explain WHY the method does better under different conditions than others, what do you see as the main impact of the results? What are the key limitations? Some of the introduction discussion of limitations could be brought in here and discussed within the context of the results found

**Authors:** We respectfully disagree with the Editor's opinion. All the attention given to the site description (section 2.3) is of fundamental importance for understanding the limitations of the current validation and to interpret the results. Here we have about 4 pages of quantitative results. The results section covers another five pages, not 17 lines. All results reported in the figures have been commented, while trying to provide plausible explanations. As also stated in the answer to the Editor's comment 2, we do not believe that the scientific value of the results should be based on the number of pages of the "Results" section. In our opinion, explanations that are clear and to the point are more effective than excessively long and verbose ones. For the same reason, we do not think it is necessary to further expand this section just to repeat what was discussed in detail in previous parts of the manuscript.

35. **Editor:** Tying the results back to the underlying physical phenomena, models, experimental set up and the triangulation of the 3 to explain what you understand and what the study tells you is critical to establishing the scientific value of the paper.

Authors: We agree, this is exactly what is done in the "Result" section.

Just to make an example, we are reporting the discussion of Fig. 9, at page 15 lines 22-27: "The top plot of the figure shows the lower-half-rotor shears measured at the met-mast and by the sector-equivalent speeds. Although some discrepancies are present, the figure shows that the sector-effective observer is capable of following the main changes in shear captured by the met-mast. The main discrepancies can be found between 2PM of October 21 and about 4AM of October 22, when WT1 is in the wake of WT2 or in its close proximity. However, one should not forget that the two estimates correspond to two locations spaced 2.5D apart, and that the exact ground truth at the rotor disk —where the observers operate— is unknown."

In the example, we are clearly characterizing the results at the light of the physical phenomena, while taking into account the limits of the available experimental setup.

**Conclusions**

36. **Editor:** *Revisit the conclusions once the rest of the paper updates are made. A lot of the previous comments also apply here.*

**Authors:** It is hard to understand to which of the previous comments the reviewer is referring to. We believe the reviewer might find the conclusions too vague. The text of this section has been modified to more precisely summarize the observer performance.

**37. Editor: *Strengthen the overall closing statements* Authors: The text has been revised.**

We have taken the opportunity to make several small editorial changes to the text, in order to improve readability. A revised version of the manuscript is attached to the present reply, with the main additions highlighted in blue and deletions in red.

Best regards, The authors

[revised manuscript text omitted]
 m is given by Eq. (4) and r is the residual with covariance  $Q = \mathbf{E}[rr^T]$ . Residuals are assumed to be zero-mean and colored, and are due to measurement noise and unmodeled physics (Jategaonkar, 2015). Given the model coefficients, a maximum likelihood (Strutz, 2016) estimate  $\theta_{\rm E}$  of the wind states can be computed online during turbine operation from the measured loads  $m_{\rm M}$  from Eqs. (4) and (8) as follows

$$\boldsymbol{\theta}_{\mathrm{E}} = \left(\boldsymbol{F}^{T}\boldsymbol{Q}^{-1}\boldsymbol{F}\right)^{-1}\boldsymbol{F}^{T}\boldsymbol{Q}^{-1}(\boldsymbol{m}_{\mathrm{M}} - \boldsymbol{m}_{0}). \tag{9}$$

**2.2.1 Density correction**

5

10

Aerodynamic loads moments can be written as

$$m_{\rm A} = q\underline{AARC}(\underline{V}, \underline{\rho}),\tag{10}$$

where  $q = 1/2\rho V^2$  is the dynamic pressure,  $A = \pi R^2$  is the rotor disk areaand, while C is a non-dimensional coefficient. A correction for density can be simply obtained as

$$m_{\rm A_{\rm ref}} = m_{\rm A_i} \frac{\rho_{\rm ref}}{\rho_i},\tag{11}$$

[revised manuscript text omitted]